# Effects of speech periodicity and speech rate on auditory-motor coupling during speech comprehension
Sojeong Kwon [1,2,4] ✉, Christina Lubinus[1,2,4], Christian A. Kell [2], Anne Keitel [3,5] &
Johanna M. Rimmele [1,2,5] ✉

According to neural oscillatory accounts, periodicity at the syllabic scale enhances speech comprehension through theta brain rhythms. Natural speech, however, is not strictly periodic and stronger periodicity, such as under conditions of fast speech, may hinder comprehension. Using magnetoencephalography, we investigate how natural variation in syllabic-level periodicity affects comprehension and auditory-motor coupling in brain areas related to temporal speech processing. We model speech periodicity and rate independently. Theta-band phase coupling between the posterior superior temporal gyrus (pSTG) and speech motor areas is assessed using Gaussian-Copula Mutual Information (GCMI). We find that faster syllabic rates and lower periodicity are associated with stronger coupling between the pSTG and inferior precentral gyrus, but also inferior frontal gyrus and supplementary motor areas. Comprehension improves with lower periodicity and declines at faster rates. The syllabic rate and periodicity moderate the coupling-comprehension relationship, possibly reflecting a complex interplay of lower-level auditory processing and higher-level prediction from the speech motor cortices. These findings suggest a sweet spot for natural, less periodic speech rhythms that support optimal processing and emphasize the necessity to investigate natural speech.

Temporal processing is crucial for speech processing and comprehension[1–5]. Speech typically follows a quasi-periodic rhythm at the syllabic time scale[6]. Rhythmicity appears to be an important feature and rhythmic qualities of speech may vary depending on the content[7], the speaker[8], the language[9,10], or the speed of speaking[11,12].

Empirical results on whether periodicity at the syllabic time scale has beneficial effects on comprehension are mixed. Some behavioral research suggests that a regular timing of stressed vowels in disyllabic words improved spoken-word perception[13]. Furthermore, for people with aphasia, speech isochrony at the syllabic level enhances neural speech tracking and supports comprehension[14]. In studies that re-timed speech units, periodicity compared to random timing at the syllabic level improved speech perception in noise[15,16]. However, natural non-periodic speech was shown to be more intelligible than isochronously retimed speech[15,16]. Aubanel and Schwartz[16] concluded that predictive cues related to bottom-up isochrony and top-down naturalness may be combined to aid speech comprehension. In summary, there are controversial findings on whether speech comprehension benefits from periodicity. Crucially, natural speech is not strictly periodic and there is a lack of research examining how natural variation in syllabic-level periodicity affects speech comprehension and neural processing.

An influential neural oscillatory account emphasizes how cortical oscillations operate upon syllabic-level periodicity to facilitate speech comprehension[17–21]. According to this, populations of neural theta-band (4-8 Hz) oscillations in auditory cortex align their high excitability phases[21,22] to the slow temporal fluctuations in the acoustic speech envelope at the syllabic scale to aid syllabic segmentation, a basis for comprehension[6,19,23–27]. Because the involvement of neural oscillations is typically not directly shown, we use the descriptive term *speech tracking* when reporting research on the alignment of neural activity to the speech acoustics. The neural oscillatory account assumes stimulus periodicity to be a relevant feature, with higher degrees of periodicity being expected to increase the entrainment strength and subsequently comprehension[28,29]. Additionally, to such rhythm processing in the auditory cortex, predictions from the motor cortex may be

[1]Department of Cognitive Neuropsychology, Max Planck Institute for Empirical Aesthetics, Frankfurt am Main, Germany. [2]Cooperative Brain Imaging Center, Goethe University Frankfurt, Frankfurt am Main, Germany. [3]Psychology Division, University of Dundee, Dundee, UK. [4]These authors contributed equally: Sojeong Kwon, Christina Lubinus.[5]These authors jointly supervised this work: Anne Keitel, Johanna M. Rimmele. ✉e-mail: sojeong.KWON@univ-amu.fr; johanna.rimmele@ae.mpg.de

particularly relevant for processing the complex temporal structure of speech[30–33]. Thereby, predictions may vary in strength for periodic and non-periodic speech, as features, such as prosody or higher-level linguistic processing, affect the temporal predictability of speech[7,11,12]. Interestingly, temporal predictions generated in speech-motor areas about upcoming sensory events during speech perception have also been linked to neural oscillations[30,34–36]. Evidence for bi-directional theta-theta auditory-motor phase coupling has been provided[37–39], with the top-down coupling directly modulating the speech tracking in auditory cortex[39]. Several spectral channels for predictive frontal-motor top-down effects during speech comprehension have been proposed: Two possibilities are phase coupling within the delta-theta band from frontal-motor to temporal cortex and between the beta and delta-theta band to primary auditory cortex. Effects often have been claimed in the delta band, but Park and colleagues suggested that it depends on the stimulus material whether effects in the delta or theta band are observed[35,39] (see also[31]). Endogenous beta and delta-theta brain rhythms have been typically observed in speech motor cortices during rest[40,41] and during language comprehension[42–44], possibly supporting timing and speech planning through coupling with the auditory cortex[40,42,45]. While delta-theta to beta phase-amplitude coupling has been related to top-down temporal predictions from the speech motor cortex, delta-theta to gamma phase-amplitude coupling has been related to simpler bottom-up driven temporal predictions[46–48]. In summary, according to the neural oscillatory account, we would expect higher speech periodicity to not only be beneficial for comprehension but also modulate auditory-motor coupling.

Regarding anatomical areas, speech perception as a multi-stage process involves various brain regions that interact during auditory-motor coupling to extract meaningful information from sound waves. A close relationship between temporal predictions and the motor system has been proposed, underscoring the dorsal stream's role in handling temporal aspects of speech[49,50]. After initial processing in subcortical areas and the primary auditory cortex[26,51–53], the auditory association cortex, particularly the posterior superior temporal gyrus (pSTG)[54,55], supports phonological processing. A dorsal ("where" or "how") stream connects the pSTG with Broca's area[56–58] and other motor-related regions essential for auditory-motor integration[59–61]. Regions like the inferior frontal gyrus (IFG), precentral gyrus (PCG), and supplementary motor area (SMA) play a role not only during production but also the perception and comprehension of spoken language[46,62–71]. fMRI research suggests that both the SMA and IFG also play a role in processing syllabic-level periodicity, when comparing isochronous and non-isochronous pseudo-sentences[35,72,73]. Outside of speech processing, listening to more regular beats[47,48], tones[74], and musical notes[75] has been shown to increase activation in SMA and PCG. This suggests that speech motor regions may be sensitive to the degree of speech periodicity at the syllabic scale, whereas the specific function and neural mechanisms are not well understood.

In the current MEG study, we investigate how natural variations in speech periodicity at the syllabic level modulate the theta phase coupling between auditory (pSTG) and speech-motor areas (IFG, inferior PCG [iPCG], and SMA) and comprehension performance during a speech repetition task (Fig. 1a). We increase variability in periodicity by studying fast in addition to slow speaking conditions and thus focus on the natural variations in speech rate. Additionally, we used compression to generate syllabic rate conditions (5–17.5 Syl/s) and to control for the effects of speech rate. These manipulations allowed us to disentangle the effects of speech periodicity and syllabic rate, which are naturally intertwined. Our findings suggest that natural, less periodic speech rhythms are optimally processed, with lower periodicity being associated with stronger auditory-motor coupling and with higher comprehension. Particularly, the coupling between the pSTG and iPCG, but also pSTG and IFG and SMA, was sensitive to periodicity. Auditory-motor coupling predicted comprehension in a complex relationship that was moderated by the syllabic rate and periodicity of

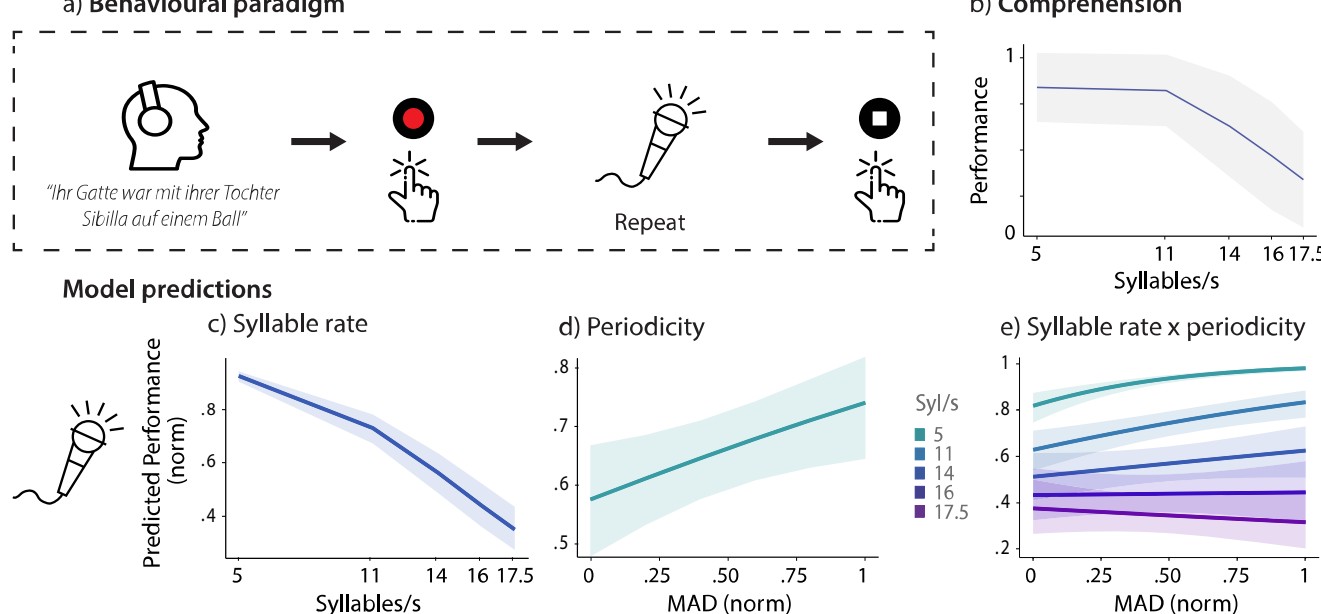

**Fig. 1 | Behaviora paradigm and periodicity effects on comprehension. a** The study design schematic illustrates a single trial from the comprehension task. In each trial, participants listened to a sentence (example here: "Her husband went to a ball with her daughter Sibilla"), then pressed a recording button to verbally repeat the sentence as accurately as possible. Once they completed their recall, they stopped the recording with another button press. A total of 300 sentences were presented, divided into blocks of 30 sentences each, with self-paced breaks between blocks. **b** The raw data indicating the performance in the comprehension task across various syllabic rates is displayed. The dark solid line represents participants' average performance, while the gray shaded area illustrates the standard error. **c** The fixed effect of syllabic rate on comprehension from the generalized linear mixed model (GLMM) analysis is shown. Comprehension decreased as the syllabic rate increased. **d** The effect of periodicity (median absolute deviation, MAD) on comprehension is displayed. With increasing MAD comprehension performance improved. **e** The interaction effect of periodicity and syllabic rate is shown. Positive slopes were observed at all syllabic rates but 17.5 Syl/s. Reduced periodicity (i.e., increased MAD) was generally associated with improved performance, particularly at slower rates. All icons, the "Click" by Wahyu Prihantoro, the "Brain" by Lewen Design, the "Head" by Ainul Muttaqin and the "Podcast" by Srinivas Agra, were used under the Creative Commons Attribution 3.0 license via *The Noun Project*.

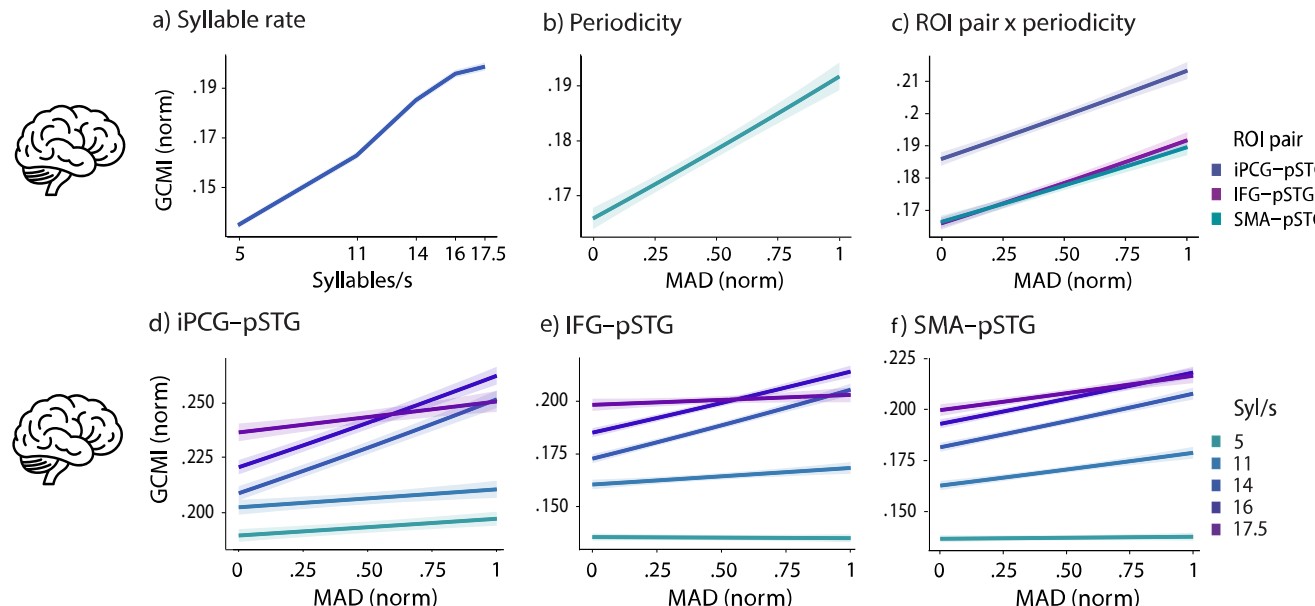

**Fig. 2 | Periodicity and Syllabic Rate affect Auditory-Motor Cortex Coupling.**
**a** The fixed effect of syllabic rate on coupling (normalized GCMI) observed in the GLMM is displayed. Coupling increased with higher syllabic rates. **b** The figure shows the fixed effect of periodicity (MAD) on coupling. Coupling increased as MAD increased (indicating lower periodicity). **c** The interaction between periodicity and ROI Pairs is shown. The iPCG–pSTG pair exhibits the highest GCMI increases with lower periodicity (higher MAD) followed by IFG–pSTG and SMA–pSTG. A positive effect of MAD reflects a negative effect of periodicity on neural coupling. Other interaction effects between syllabic rate, MAD, and brain area pairs were also evident, but are not displayed here. ROI specific analysis: **d** A iPCG-pSTG GLMM

analysis shows coupling (GCMI) strengthens with higher syllabic rates and greater MAD values (lower periodicity). A significant interaction effect between periodicity and syllabic rate is observed. **e** IFG-pSTG GLMM analysis: Coupling increases with higher syllabic rates and lower periodicity. Reduced coupling is consistently seen at lower rates, with a significant interaction effect between periodicity and syllabic rate. **f** SMA-pSTG GLMM analysis: GCMI rises with higher syllabic rates and lower periodicity. Lower rates are linked to weaker coupling, and although the interaction effect is less pronounced, it remains statistically significant. The "Brain" icon by Lewen Design was used under the Creative Commons Attribution 3.0 license via *The Noun Project*.

speech. The findings are in line with a sweet spot for speech processing of natural, less periodic speech.

## Results
### Periodicity and syllabic rate affect comprehension
We investigated 57 participants who were asked to repeat auditory sentences (Fig. 1a). In our study, we aimed to disentangle the naturally correlated factors of syllabic rate and periodicity by study design; however, multi-collinearity issues persisted. To address this, the syllabic rate was included as a predictor in the analysis, and the periodicity and syllabic rate were decorrelated using the residuals of our measure of aperiodicity (the Median Absolute Deviation, MAD) calculated within each syllabic rate group.

For the comprehension model (n = 57), a generalized linear mixed model (GLMM; Full Statistics in Supplementary Table 1) indicated a strong fit ($Rm^2 = 0.503$, $Rc^2 = 0.999$). As visible in the raw performance data (Fig. 1b), we found a fixed effect of syllabic rate on comprehension, where comprehension performance declined at higher syllable rates ($\beta = -0.16$, SE = 0.02, p < 0.001; Fig. 1c). A significant fixed effect of periodicity was found, indicating that increased MAD (reflecting reduced periodicity) was associated with improved comprehension performance ($\beta = 3.41$, SE = 0.051, p < 0.001; Fig. 1d). Furthermore, we observed a significant interaction between periodicity and syllabic rate ($\beta = -0.21$, SE = 0.04, p < 0.001, Fig. 1e). At lower syllabic rates lower periodicity (higher MAD) was related to higher comprehension performance. At higher syllabic rates, the influence of periodicity was weaker, suggesting that periodicity becomes less important as the syllabic rate increases. The analysis also revealed significant effects for control variables. The number of syllables in a sentence significantly influenced comprehension performance ($\beta = -0.38$, SE = 0.04, p < 0.001), as did the compression factor ($\beta = 0.22$, SE = 0.03, p < 0.001). However, sentence-level average word-frequency, which was used as a measure of word complexity, did not show a significant effect ($\beta = 0.02$,

SE = 0.04, p = 0.704). In summary, the results highlight that slower syllable rates and reduced periodicity (higher MAD values) are associated with better comprehension performance.

### Periodicity and syllabic rate affect auditory-motor cortex coupling across ROI pairs
The auditory-motor coupling across frequencies (Gaussian-Copula Mutual Information, GCMI) showed 1/f dynamics with a "bump" in the theta range (4–8 Hz; Supplementary Figs. 1 and 2). Based on theoretical considerations and visual inspection, the theta band, which has been previously linked to auditory-motor coupling, was chosen for further analysis, with GCMI values averaged across 4–8 Hz.

To assess the effects of periodicity and syllabic rate on auditory-motor coupling, a GLMM was employed (n = 57), predicting coupling (GCMI) based on syllabic rate, periodicity (MAD), ROI, hemisphere and their interactions, as well as several control variables (e.g., compression factor, sentence-averaged word frequency) (Fig. 2). The GLMM model explained a moderate proportion of the variance in coupling ($Rm^2$: 0.161, $Rc^2$: 0.170). Both fixed effects of syllabic rate (linear: $\beta = 146.30$, SE = 2.84, p < 0.001, quadratic: $\beta = 12.35$, SE = 3.10, p < 0.001, and cubic: $\beta = 9.37$, SE = 3.23, p = 0.006; Fig. 2a) and periodicity ($\beta = 0.10$, SE = 0.01, p < 0.001; Fig. 2b) were statistically significant, as was the effect of compression factor ($\beta = 0.01$, SE = 0.001, p < 0.001), hemisphere ($\beta = -0.08$, SE = 0.001, p < 0.001), and word frequency ($\beta = -0.004$, SE = 0.005, p < 0.001). However, no significant interaction of hemisphere and periodicity was observed (b = -0.01, SE = 0.01, p = 0.385). Coupling increased with lower periodicity, with higher syllabic rate and decreased for the right hemisphere. A fixed effect of ROI was observed (iPCG-pSTG: $\beta = 0.16$, SE = 0.004, p < 0.001, SMA-pSTG: $\beta = -0.01$, SE = 0.004, p = 0.002). Post-hoc comparisons revealed significant differences in coupling (GCMI) between the iPCG-pSTG and SMA-pSTG, as well as between the iPCG-pSTG and IFG-pSTG

https://doi.org/10.1038/s42003-025-09481-y                                                                                    **Article**

($p < 0.001$). However, no significant difference was found between the SMA-pSTG and IFG-pSTG ($p = 0.61$). Among these pairs, the iPCG-pSTG exhibited the strongest coupling, followed by the IFG-pSTG and SMA-pSTG pairs. A significant interaction effect of periodicity (MAD) and ROI was observed (Fig. 2c) (MAD effects differed between IFG-pSTG and iPCG-pSTG: $\beta = 0.02$, SE = 0.009, $p = 0.035$; but not between IFG-pSTG and SMA-pSTG: $\beta = 0.01$, SE = 0.01, $p = 0.29$). The interaction was further explored through post-hoc tests comparing the slopes for the ROI pairs, which were as follows: iPCG-pSTG ($\beta = 0.177$), IFG-pSTG ($\beta = 0.173$), and SMA-pSTG ($\beta = 0.159$). In the post-hoc tests, however, across periodicity conditions no significant differences between ROI pairs were shown (iPCG-pSTG and IFG-pSTG: $p = 0.79$; iPCG-pSTG and SMA-pSTG: $p = 0.62$; IFG-pSTG and SMA-pSTG: $p = 0.62$). This suggests that periodicity effects did not strongly differ across ROIs. Furthermore, significant interaction effects were identified between syllabic rate and periodicity, periodicity and ROI pairs, and syllabic rate and ROI pairs, as well as the combined three-way interaction of syllabic rate, periodicity, and ROI pairs (for detailed statistics, see Supplementary Table 2). These interactions are further explored below.

### ROI specific analyses of periodicity and syllabic rate effects

To further explore the interaction effects between syllabic rate and periodicity, periodicity and ROI pairs, and syllabic rate and ROI pairs, as well as the combined three-way interaction of syllabic rate, periodicity, and ROI pairs (for detailed statistics, see Supplementary Table 2), separate analyses of periodicity and syllabic rate effects were conducted for each ROI.

**iPCG-pSTG pair.** For the iPCG-pSTG pair (Fig. 2d), the GLMM model (n = 57) accounted for a low-moderate proportion of the variance ($Rm^2$: 0.100, $Rc^2$: 0.124). Both main effects of periodicity ($\beta = 0.13$, SE = 0.01, $p < 0.001$) and syllabic rate were significant (linear: $\beta = 53.97$, SE = 1.62, $p < 0.001$, quadratic: $\beta = 17.75$, SE = 1.81, $p < 0.001$, and cubic: $\beta = 9.11$, SE = 1.88, $p < 0.001$), along with a significant interaction effect between them (linear: $\beta = 22.44$, SE = 3.44, $p < 0.001$, quadratic: $\beta = -14.23$, SE = 3.93, $p = 0.001$, and cubic: $\beta = -42.01$, SE = 4.13, $p < 0.001$). Coupling increased with both higher syllabic rates and lower periodicity (MAD), with the steepest slopes observed at 14 Syl/s (slope: 0.242) and 16 Syl/s (slope: 0.228) (Fig. 2d). In contrast, the flattest slopes were found at 5 Syl/s (slope: 0.049), followed by 11 Syl/s (slope: 0.050) and 17.5 Syl/s (slope: 0.076). Pairwise comparisons revealed significant differences in slope steepness between 5 Syl/s, 11 Syl/s, and 17.5 Syl/s versus 14 Syl/s and 16 Syl/s ($p < 0.001$). However, no significant differences were found among 5, 11, and 17.5 Syl/s or between 14 and 16 Syl/s (all p-values > 0.14). These findings suggest that sensitivity to periodicity was most pronounced at mid-range syllabic rates, specifically between 14 and 16 Syl/s.

**IFG-pSTG Pair.** The analysis for the IFG-pSTG pair (Fig. 2e; n = 57) revealed a moderate proportion of variance explained ($Rm^2$: 0.150, $Rc^2$: 0.161). As with the iPCG-pSTG pair, the main effects of both periodicity ($\beta = 0.10$, SE = 0.01, $p < 0.001$) and syllabic rate (linear: $\beta = 86.91$, SE = 1.61, $p < 0.001$, quadratic: $\beta = 7.84$, SE = 1.77, $p < 0.001$, and cubic: $\beta = 5.73$, SE = 1.83, $p < 0.001$) were significant, as was their interaction (linear: $\beta = 22.49$, SE = 3.40, $p < 0.001$, quadratic: $\beta = -24.22$, SE = 3.86, $p < 0.001$, and cubic: $\beta = -35.05$, SE = 4.03, $p < 0.001$). The steepest slopes were observed at 14 Syl/s (slope: 0.214) and 16 Syl/s (slope: 0.182) (Fig. 2e), while smaller slopes were seen at 11 Syl/s (slope: 0.057), 17.5 Syl/s (slope: 0.029), and 5 Syl/s (slope: -0.004). Like for the iPCG-pSTG pair, the differences in slopes between the edge group (5, 11, and 17.5 Syl/s) and the mid-range group (14 and 16 Syl/s) were all significant ($p < 0.001$). However, within the edge group, slope differences were not statistically significant (all p > .11). In contrast, a significant difference was observed between 5 and 11 Syl/s ($p = 0.001$), and 14 and 16 Syl/s ($p = 0.002$). These findings, like those observed in the iPCG-pSTG pair, indicate that sensitivity to periodicity was highest at mid-range syllabic rates, 14 and 16 Syl/s.

**SMA-pSTG pair.** For the SMA-pSTG pair (Fig. 2f), the model (n = 57) accounted for a moderate proportion of the variance ($Rm^2$: 0.160, $Rc^2$: 0.170). Coupling strength increased with both syllabic rate (linear: $\beta = 91.94$, SE = 1.62, $p < 0.001$, quadratic: $\beta = 2.60$, SE = 1.78, $p = 0.142$, and cubic: $\beta = -3.94$, SE = 1.83, $p = 0.034$) and MAD ($\beta = 0.11$, SE = 0.01, $p < 0.001$). The interaction effects were significant (linear: $\beta = 22.55$, SE = 3.41, $p < 0.001$, quadratic: $\beta = -17.85$, SE = 3.86, $p < 0.001$, and cubic: $\beta = -10.69$, SE = 4.03, $p = 0.009$). Sensitivity to periodicity was most prominent at 14 Syl/s (slope: 0.168), with moderate slopes observed at 16 Syl/s (slope: 0.154), 11 Syl/s (slope: 0.113) and 17.5 Syl/s (slope: 0.102), and lower sensitivity at 5 Syl/s (slope: 0.008). Pairwise comparisons confirm these findings: the slope at 5 Syl/s was significantly flatter compared to all other syllabic rates ($p < 0.001$). Significant differences were observed between 17.5 Syl/s and both 14 Syl/s ($p = 0.008$) and 16 Syl/s ($p = 0.005$), as well as between 11 Syl/s and both 14 Syl/s ($p = 0.005$). However, other comparisons were not statistically significant, including 11 vs. 16 Syl/s, 11 vs. 17.5 Syl/s and 14 vs. 16 Syl/s (all p-values > 0.06). These results suggest that for the SMA-pSTG pair, apart from the notably flatter slope at 5 Syl/s, the slopes for other syllabic rates were relatively similar. Unlike the iPCG-pSTG and IFG-pSTG pairs, there was no distinct steepness for the 14 and 16 Syl/s slopes, highlighting a different pattern of sensitivity to periodicity.

In summary, across ROI pairs, both higher syllabic rates and lower periodicity were associated with increased coupling strength. Overall, the iPCG-pSTG pair exhibited the highest coupling, followed by the IFG-pSTG and SMA-pSTG pairs. The sensitivity to periodicity (indicated by the steepness of slopes) slightly differed across ROI pairs, with higher sensitivity for iPCG-pSTG vs. SMA-pSTG coupling. These findings suggest that lower periodicity is related to higher auditory-speech-motor coupling within natural speech rhythms.

### Periodicity and syllabic rate moderate the coupling-comprehension relationship

In a direct analysis (GLMM, n = 57) of the relationship of speech comprehension and auditory-motor coupling, we predict comprehension by the auditory-motor coupling (GCMI), the periodicity, the syllabic rate, as well as the interactions between coupling periodicity and syllabic rate (Fig. 3). The interactions test whether the relationship between comprehension and coupling (i.e., the slope) is moderated by the syllabic rate or periodicity. A fixed effect of GCMI ($\beta = 3.46$, SE = 0.15, $p < 0.001$) suggests that lower comprehension was related to higher coupling. Additionally, two-way interactions of GCMI and syllabic rate ($\beta = -0.25$, SE = 0.01, $p < 0.001$) and GCMI and speech periodicity ($\beta = -3.36$, SE = 0.31, $p < 0.001$) were observed. Furthermore, the effects were specified by a three-way interaction ($\beta = 0.19$, SE = 0.02, $p < 0.001$), suggesting that the relationship between comprehension and coupling is jointly moderated by syllabic rate and periodicity. Thereby, periodicity seems to particularly impact the effect of coupling on comprehension at slower syllabic rates (5 syl/sec, 11 syl/sec), with higher coupling related to higher comprehension for the high periodicity speech, whereas for the low periodicity speech, no strong comprehension-coupling relationship seems present at slow syllabic rates. At higher syllabic rates for all periodicity conditions, low coupling was related to high comprehension. The model accounted for a high proportion of the variance ($Rm^2$: 0.585, $Rc^2$: 0.790).

### Discussion

In this study, we investigated the effects of natural variations in syllabic-level periodicity on comprehension and auditory-motor coupling strength. Our findings show that, under conditions of lower compared to higher periodicity, comprehension is better and neural coupling between auditory and speech motor brain areas is stronger. Although the auditory-motor coupling strength varied across brain areas, all regions showed sensitivity to periodicity. Our findings shed new light on the role of speech periodicity for comprehension, suggesting a sweet spot for the comprehension of natural, low periodic syllabic rates. Moreover, speech motor areas were most

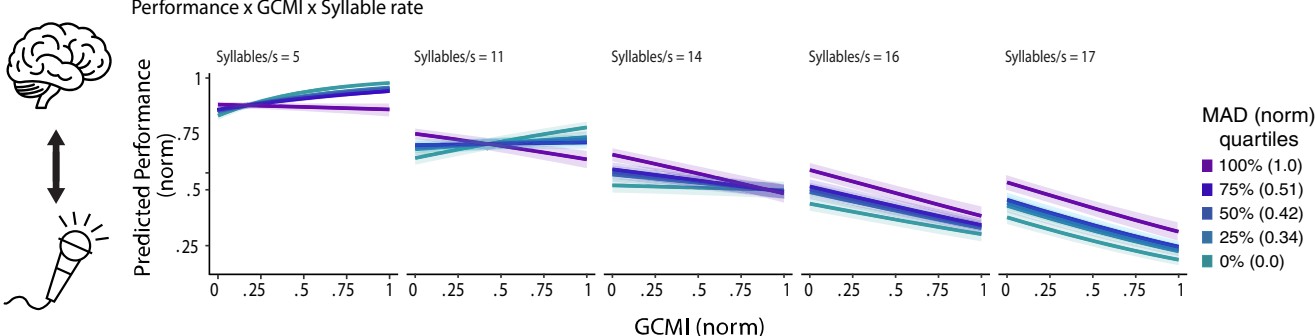

**Fig. 3 | Speech periodicity and syllabic rate moderate the coupling-comprehension relationship.** The figure displays how the three-way interaction between auditory–motor coupling (GCMI), syllabic rates, and periodicity predicts speech comprehension. Lines within each panel represent quartiles of MAD, ranging from low MAD (high periodicity) to high MAD (low periodicity). At slower syllabic rates (5 Syl/s, 11 Syl/sec) for higher periodicity (low MAD) speech, higher auditory-

motor coupling (GCMI) was related to higher comprehension. Less of a relationship seems evident for low periodicity speech (at 5 Syl/s). At higher syllabic rates, higher comprehension was related to lower GCMI. The "Brain" icon by Lewen Design and the "Podcast" by Srinivas Agra were used under the Creative Commons Attribution 3.0 license via *The Noun Project*.

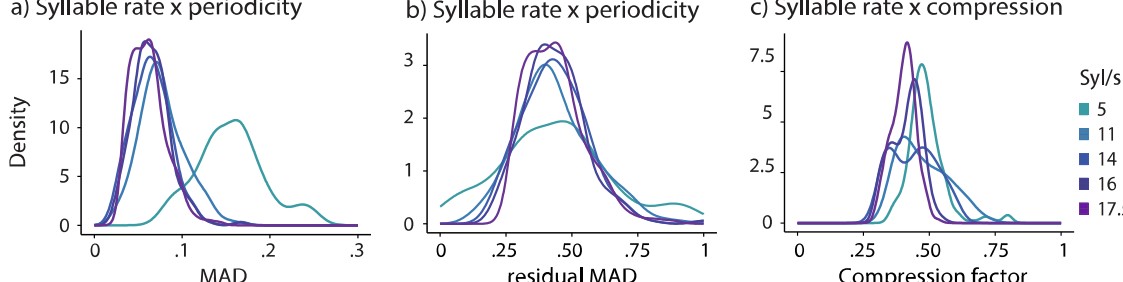

**Fig. 4 | Periodicity and compression characteristica. a** The median absolute deviation (MAD) distribution plot across different syllabic rate conditions. The distribution peak at 5 Syl/s is notably shifted to the right compared to the other conditions. **b** The distribution of residual MAD across syllabic rates, normalized to

minimum and maximum values, is shown. The residual MAD across conditions was aligned, which is utilized for statistical analysis. **c** Density plot illustrating the compression factor for each syllabic rate condition. The differences in compression factors across the conditions were minimized.

strongly recruited in this natural setting. Additionally, we disentangled the effects of speech periodicity from that of syllabic rate, which are naturally correlated[11]. We found that, at faster syllabic rates, comprehension performance is lower while auditory-motor coupling strength increases in such demanding listening situations (across speech motor regions of interest). We observed a complex relationship between auditory-motor coupling and comprehension that was moderated by the syllabic rate and periodicity of speech. These results possibly reflect both: increased temporal predictions from the motor system to facilitate comprehension in demanding listening situations and optimal auditory cortex responsiveness to natural low periodicity speech. Our findings indicate that listening to more periodic speech does not engage auditory-motor systems in the same way as less periodic speech. This might have implications for research using isochronous speech. In contrast to natural speech, isochronous speech may particularly rely on non-optimal rhythm processing in the auditory cortex and differently engage temporal predictions from the motor system[36].

Speech comprehension performance decreased with higher syllabic-level periodicity. In the context of neural oscillatory accounts, we would have expected that high periodicity is beneficial for comprehension, given the expected better neural oscillatory alignment to the speech acoustics aiding syllabic-level temporal processing (segmentation) and subsequently improved phoneme encoding[25,76–78]. In line with our finding, however, several previous studies find that while syllabic rate isochronicity is advantageous compared to a random structure, it can reduce comprehension compared to natural non-periodic rhythmicity found in speech[14–16]. In contrast, in our study, natural degrees of periodicity were investigated based on slower and faster speaking rates, with faster speaking rates being related to higher degrees of syllabic-level periodicity (Fig. 4a). The higher degree of

periodicity in fast compared to slower speech has been reported previously[11,12]. This finding suggests that for speech comprehension, a relatively low degree of syllabic-level periodicity, as found in natural speech spoken at a comfortable rate, may be beneficial. The temporal structure, such as the prosody of natural non-periodic speech, may carry relevant information at various linguistic levels that is removed in artificial isochronous speech[16], but likely also reduced in naturally more periodic, faster speech. A possibility is that the increased comprehension is related to the auditory cortex being optimally responsive to natural low periodicity speech, as suggested by computational modeling work[79]. We tested, furthermore, whether the periodicity effects on comprehension were reflecting effects of sentence-level average word-frequency, as an estimate of word complexity. When speech is produced, word frequency may impact the periodicity of the speech signal[7,16], with more frequent words being preceded by shorter words[7]. In our study, however, the sentence-level average word frequency did not predict speech comprehension, while it did predict auditory-motor coupling. Importantly, periodicity effects were observed when word frequency was controlled for.

On the neural level, we found lower syllabic-level periodicity to be associated with stronger theta-band auditory-motor phase coupling. Notably, the overall coupling strength varied across ROI pairs. Furthermore, the periodicity effects slightly varied across ROIs. Our analyses focused on the theta-band, as prior research indicates the theta band's relevance in the bi-directional theta-theta auditory-motor phase coupling while listening to syllables and continuous speech[35,37–39,80]. In line with this prior research, we observed the strongest auditory-motor phase coupling in the theta-band (knee bump at 4–8 Hz; Supplementary Figs. 1 and 2). However, although speech processing likely involves auditory-motor theta-band coupling, the

spectral interactions involved in speech processing are likely more complex. In our study, periodicity affected theta-band auditory-motor phase coupling across all ROI pairs, with slight differences in sensitivity across ROIs (significant interaction), suggesting weaker sensitivity observed in the SMA-pSTG pair compared to the iPCG-pSTG pair (Fig. 2). Speech-motor areas, including the SMA, have been shown to be activated during listening to speech[46,62-71]. Particularly, the SMA (i.e. SMA proper[81-83]) has been reported to play a role in rhythmic processing, as it is activated when listening to regular beats[47,48], tones[74], and musical notes[75]. One possible explanation for our contradictory finding is that the processing of speech rhythm differs from that of rhythm and beat in musical stimuli (or pure tone sequences), possibly due to the recruitment of speech-specific processing routes that differentially affect SMA responsiveness. Another possibly related factor could be the relatively high-frequency range of syllabic rates used in this study (5–17.5 Syl/s), while SMA is typically activated by slower rhythms below 5 Syl/s[47,48,74].

We observed that comprehension performance declined at higher syllabic rates. This effect aligns with previous research emphasizing the role of speech rate in comprehension, where higher speech rates are associated with more difficult listening situations and lower comprehension performance[1,6,43,84-86]. At the neural level, theta-band auditory-motor phase coupling increased with higher syllabic rates, being lowest at 5 Syl/s (Fig. 2a). Previous research has suggested that auditory-motor coupling might decrease at higher rates, with coupling being highest around 5 Syl/s[38]. However, the speech rates included in this study were much lower than in our study. In our study, speech motor areas were particularly recruited at high-demanding speech rates. This is in line with previous findings that the motor system is particularly recruited in demanding listening situations, such as listening to speech-in-noise[70,87].

Interestingly, a direct analysis revealed a complex relationship between speech comprehension performance and auditory-motor coupling that was jointly moderated by the syllabic rate and the speech periodicity (Supplementary Fig. 3). A fixed effect suggests that lower comprehension was related to higher auditory-motor coupling. This relationship, however, was further specified by two-way and three-way interactions. Both high and low auditory-motor coupling could predict high comprehension, depending on the periodicity and syllabic rate context. We speculate that the findings can be accounted for by a complex interplay of lower-level temporal processing in auditory cortex and higher-level temporal prediction from the speech motor cortices, possibly related to linguistic and prosodic information[33]. More specifically, at slower syllabic rates, the motor cortex may provide temporal predictions of the speech signal. In the case of high periodicity speech, such temporal motor predictions may be useful, as the periodic signal lacks the natural temporal structure of speech. This possibly resulted in higher auditory-motor coupling being related to higher comprehension when syllabic rates are slow, and periodicity is high. In the case of low periodicity speech presented at slow syllabic rates, auditory cortex processing may be optimal, given that the natural temporal structure of speech is preserved, and individuals may more strongly rely on sensory representations. For fast syllabic rates, given the demanding listening situation, temporal predictions from the motor cortex may be expected to be particularly useful, as shown for simple tone sequences[88]. Others, however, did not find increased auditory-motor engagement at fast speech rates[85]. We find that for fast speech rates, higher comprehension was related to lower coupling. A possibility is that because of our speech production experience, the speech motor cortex provides exact temporal prediction for speech spoken at a natural, slow tempo. At very fast rates outside of our production abilities, in contrast, predictions may be detrimental. Three-way interactions, however, should be interpreted cautiously, and further research should explore this complex comprehension-coupling relationship. It is possible that, due to the requirement of verbally repeating the sentence, our speech comprehension task emphasized the auditory-motor coupling aspect. However, the engagement of the motor system has been shown even during passive listening to speech[38,39], and similar behavioral effects of the motor system on speech comprehension performance have been shown with tasks that do not

involve overt speech production[41,85]. In summary, our findings suggest a sweet spot with the highest comprehension for low periodicity speech. This likely involves both processing within the auditory cortex and auditory-motor coupling, facilitating comprehension in temporally impoverished listening situations (high periodicity speech). Our findings indicate that listening to more periodic speech does not engage auditory-motor systems in the same way as less periodic speech. This might have implications for research using isochronous speech. In contrast to natural speech, isochronous speech may particularly rely on non-optimal rhythm processing in the auditory cortex and differently engage temporal predictions from the motor system[36].

In this study, syllabic rate and periodicity were effectively disentangled through both the study design and the analysis of residuals (Fig. 4a, b), allowing a focused investigation of the impact of periodicity on speech comprehension and neural coupling. The design further enabled the examination of natural variability in speech rates, as speakers produced both slower and faster speech samples. To minimize potential confounds, several acoustic factors, such as syllable number, average sentence word frequency and compression, were included as control variables. Importantly, while some control variables did exhibit an effect, the primary periodicity-related effects were consistently observed. A limitation is that the mixed models in this study account for relatively low variance, and the effect sizes observed are modest, which, however, is consistent with expectations for studies of naturalistic speech, where complex stimuli typically yield subtle effects[89,90].

In conclusion, our findings demonstrate that lower syllabic-level periodicity is linked to increased theta-band phase coupling strength between auditory and speech motor regions, as well as enhanced speech comprehension, with a complex relationship between comprehension and coupling. These results underscore the relevance of natural, flexible rhythmic structures in speech for facilitating optimal comprehension.

## Methods and materials

Magnetoencephalography (MEG) data were collected in a previous study[91]. The experiment consisted of three separate sessions: a behavioral session, an MEG session (including behavioral recordings), and a structural MRI (Magnetic Resonance Imaging) session. In the current study, we only included data from the MEG and MRI sessions. The MEG data reported here were recorded during the performance of a speech comprehension (intelligibility) task. A structural MRI session was performed to gather anatomical data.

### Participants

The study analyzed data from a final sample of 57 participants[91], all of whom completed every condition of the experiment under the same procedure. Initially, 60 individuals (age: M = 26.9 years, SD = 5.4 years; 32 females, 28 males, based on self-reported gender) were recruited from the MPI database, drawing from the local Frankfurt community. However, three participants were excluded due to technical issues during recording or due to an average performance that was three standard deviations below the mean in the baseline speech comprehension task (5 Syl/s)[91]. All participants confirmed the absence of neurological or psychiatric disorders in a self-report assessment. Additionally, participants were required to have normal hearing and normal or corrected-to-normal vision. All participants were native German speakers and right-handed, factors which were considered to maintain consistency and minimize potential confounding variables across the study population. The study[91] was approved by the local ethics committee of the University Hospital of the Goethe-University Frankfurt (approval number: 2021-509) and conducted in accordance with the Declaration of Helsinki. All participants gave informed consent to the participation. All ethical regulations relevant to human research participants were followed.

### Experimental procedure

After setting up the electrooculogram (EOG) and electrocardiogram (ECG) channels, fiducial markers were placed at the nasion and preauricular points to monitor head position for accurate data acquisition throughout the

session. Participants were seated in the MEG for the comprehension task (Fig. 1a). The task was divided into 10 blocks, each lasting approximately 5 min. Trials were pseudorandomized across participants and arranged into blocks of 30 trials, with self-paced breaks between blocks. During each trial, a sentence was presented through EARTONE Gold 3 A earplugs, controlled by Matlab (R2017a) with the Psychtoolbox extension[92] on a Fujitsu-Technology CELSIUS R940power PC. Participants listened to each sentence and verbally repeated it as accurately as possible at the end of a trial. They were encouraged that, in case they did not fully understand a sentence, they should repeat the portions they could understand. Participants initiated and stopped recordings of the verbal response via button presses with the left or right index finger, initiating the interstimulus interval. Auditory stimuli were presented at five different syllabic rates, with each rate condition containing a unique set of 60 sentences, resulting in a total of 300 trials.

## Stimuli
The speech stimulus pool used for the comprehension task was sourced from German books (306 sentences) from zeno.org and audiobooks (138 sentences) from Librivox.org. For each stimulus set, 300 sentences were selected from this stimulus pool. The stimuli from books were recorded by three different native German speakers at the Max Planck Institute for Empirical Aesthetics. The speakers were instructed to speak slow, medium, and fast versions of the sentences (for details of the recording settings see ref. 91). In order to generate different syllabic rate conditions, ranging from 5 to 17.5 Syl/s, different compression was applied to all sentences in addition to the naturally occurring variance in speaking rate. This resulted in five syllabic rate conditions with the syllabic rate of trials varying around the mean syllabic rate of the condition (5 Syl/s, 11 Syl/s, 14 Syl/s, 16 Syl/s, 17.5 Syl/s) and with an overlapping distribution of compression factors (Fig. 4c). To generate stimulus lists for the comprehension task, sentences were pseudo-randomly selected from the pool of recordings. Notably, stimulus sets for the comprehension task (i.e., three different stimulus sets were used across participants) comprised 300 unique sentences, avoiding repetition. Time compression or expansion was executed using the Pitch Synchronous Overlap and Add (PSOLA) algorithm[93,94] within Praat (version 6.4.07), maintaining the periodic structure within each sound segment and enabling precise manipulation of the syllabic rate. Moreover, all audio files were root mean square standardized to a loudness of 69 dB.

To measure periodicity, we used R (version 4.3.3) to calculate the median absolute deviation (MAD) of the inter-syllabic-nuclei intervals for each sentence. High periodicity corresponds to small variation in the inter-syllabic-nuclei intervals (i.e., small MAD values). The method has been previously used to quantify periodicity, as it is relatively robust to outliers[95,96]. The inter-syllabic-nuclei intervals were calculated in the following way: Using Praat and a syllable nuclei detection script[97], we extracted the timing of each syllable nucleus from each sentence recording. For each sentence, the MAD of the timing of all syllable nuclei was computed, resulting in one MAD value per sentence. As the PRAAT algorithm is less accurate for fast speech rates, especially those exceeding 14 Syl/s, the original speech material prior to compression was used to compute periodicity. Using the original speech material is a feasible approach because the PSOLA method was used for compression. The PSOLA algorithm performs compression while the original periodic structure of the uncompressed speech is maintained through a pitch-synchronous, quasi-linear transformation, enabling precise adjustments to the syllabic rate (Fig. 4a). Paired t-tests revealed that MAD in the 5 Syl/s condition was always significantly different from the MAD in all other syllabic rate conditions (p < 0.001). Additionally, comparisons between 11 Syl/s and 14, 16, and 17.5 Syl/s conditions showed statistical significance (p < 0.001), as were the comparisons between 17.5 Syl/s and both 14 and 16 Syl/s (p < 0.001). Our analysis showed significant multi-collinearity between syllabic rate and periodicity (Variance Inflation Factor >5). To address this, we calculated the residual MAD for each sentence (computing the MAD across all stimuli first, then fitting models per syllabic rate group to obtain residuals) (Fig. 4b). After adjusting for residuals, no comparisons between syllabic rate condition pairs showed significant differences (p > 0.05). Thus, our periodicity measure (residual MAD) is largely independent of syllabic rate.

## Behavioral data analysis
In the context of Lubinus et al.[91], the recorded participants' responses were manually transcribed by the co-first author to produce text excerpts. The responses were compared to the original sentences using Python's built-in sequence matcher algorithm from *difflib* package to measure comprehension accuracy. This algorithm evaluates the similarity between two sequences based on the order of their elements (words). The sequence matcher produced a similarity percentage for each sentence (word-overlap performance), which was used as the comprehension performance measure. To assess the transcription error, two native German speakers manually checked 5% of all trials (15 trials per participant, including 3 from each syllabic-rate condition; 795 trials in total) by listening to the recorded responses and transcribing them. These transcriptions showed 90% word-overlap with those produced by the co-author. The word-overlap score was computed in Python (version 3.12.4) using the *difflib* package.

## Behavioral data statistics and reproducibility
The behavioral data (n = 57) were analyzed using generalized linear mixed models (GLMM). No replication of the analysis was included. Model fitting was performed using the Template Model Builder (TMB) in R, specifying the beta family. All analyses used the glmmTMB package with a beta distribution to appropriately model sentence-wise data bounded between 0 and 1. To address boundary values at 0 and 1, we adjusted the data by adding or subtracting a small value, calculated as ± (median/sample size). No multi-collinearity issues were observed. Sentence-wise speech comprehension performance (measured as % correct) was modeled as a function of syllabic rate, periodicity, compression factor, syllable number, stimulation order, and sentence-level average word-frequency. The word complexity sentence-level average word-frequency was estimated based on the average word frequency data per sentence from the Leipziger Wortschatz Corpus (Mixed-Typical 2011 dataset). The model included an interaction term between syllabic rate and periodicity to examine their combined influence on comprehension. Random effects were included to account for variability across participants and audio files. This model structure allowed us to capture both fixed and random effects while addressing the variability inherent to the experimental design. Note that all variables were treated as metrical.

Formula syntax for behavior model (Note that this is the syntax used in the software R. For a detailed explanation, refer to the R documentation):

$$Speech\ comprehension\ performance \sim syllabic\ rate * periodicity + compression\ factor$$
$$+ sentence - level\ average\ word - frequency + syllabus\ number + stimulation\ order$$
$$+ (1|trials) + (1|participants) + (0 + syllabic\ rate|subject)$$

## MEG recording
MEG data were acquired at a sampling rate of 1200 Hz using a 275-channel whole-head MEG system (Omega 2005, CTF Systems Inc.), housed within a magnetically shielded room. To enhance data quality and mitigate artifacts, online denoising (higher-order gradiometer balancing) and online low-pass filtering (cut-off: 300 Hz) were applied during data acquisition.

## MRI recording
Individual T1-weighted structural MRI scans (standard 1 mm T1-weighted MPRAGE) were acquired using a 3 Tesla scanner (2 participants scanned on a Siemens Magnetom Trio scanner; all other participants scanned on a Siemens Magnetom Prisma scanner, Siemens, Erlangen, Germany). To align the MRI and MEG data for source reconstruction, Vitamin E capsules were placed at key anatomical landmarks: the nasion and the left and right preauricular points. These landmarks were used to co-register the MRI scans with the MEG coordinate system through a semi-automated process.

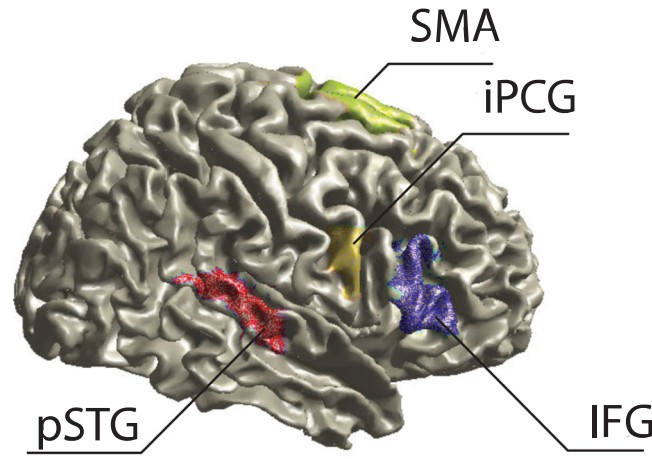

**Fig. 5 | Regions of interest (ROIs) selected from the AAL brain atlas.** ROIs are labeled with area names (note that pSTG and iPCG were modified from the original).

## MEG data analysis

The preprocessing was performed with the Fieldtrip toolbox (version 20221223[98]). A bandpass filter (0.5–160 Hz, Butterworth filter; filter order 4) was applied, followed by the removal of line noise (49.5–50.5, 99.5–100.5, 149.5–150.5 Hz, two-pass; filter order 4). A semi-automatic artifact rejection procedure was then implemented to identify jump, muscle, and threshold artifacts. For muscle artifacts, the data were filtered (110–140 Hz), while a median filter was applied for jump artifacts. Subsequently, z-transformation was performed per sensor and time point, with trials rejected if they exceeded predefined thresholds (jump: z = 45, muscle: z = 15). Threshold artifacts led to trial rejection if the range of activity at any channel surpassed a predefined threshold (threshold = 0.75e − 5). The data were down-sampled to 500 Hz and epoched (−1000ms relative to trial onset until +100 ms relative to trial offset), resulting in epochs of varying lengths. Trials containing head movements larger than 4 mm were identified and discarded using continuous fiducial measures. Subsequently, data from separate recording blocks were concatenated into one file, and sensors with high z-values computed across sensors (z > 2) were rejected. Finally, ICA uses the infomax algorithm[99] to correct eye blink, eye movement, and heartbeat artifacts. The data were further down-sampled to 100 Hz.

For source localization, T1-weighted MRIs were segmented into white matter, gray matter, and cerebrospinal fluid to create single-shell volume conduction models (head models)[100]. These images were normalized to MNI space, and individual grids with a resolution of 5 mm were generated by inverse-warping a template grid to align with each participant's anatomical structure. Using the individual grids and volume conduction models, forward models were computed to reconstruct source activity. Source reconstruction was performed using the Linearly Constrained Minimum Variance (LCMV) Beamformer method[101]. Covariance matrices were calculated across all trials, and a common spatial filter was generated using individual forward models for all experimental conditions. The lambda regularization parameter was set at 1%, and time-series data were extracted for all three dipole orientations at each voxel. Trial data were then source-localized by projecting them through the common spatial filter separately for each condition.

For the analysis of auditory-motor region coupling, Automated Anatomical Labeling (AAL) atlas-defined regions of interest (ROI) were utilized. These included pSTG and motor areas, including the IFG, iPCG, and SMA (Fig. 5). The ROIs were identified based on the AAL atlas, with specific parcels corresponding to each region extracted from the source-localized data. Original AAL ROIs' are 1 and 2 for PCG (right and left, AAL label: Precentral), 13 and 14 for IFG (right and left, AAL label: Frontal_Inf_Tri), 19 and 20 for SMA (right and left, AAL label: Supp_Motor_Area), and 81 and 82 for STG (superior temporal gyrus, right and left, AAL label: Temporal_Sup). Among these, PCG and STG underwent additional division in

order to have the inferior parts of PCG (iPCG) and the posterior parts of STG (pSTG). PCG was divided horizontally into three equal parts at the one-third and two-thirds marks, and STG was divided in half along the vertical midline of the voxel. We focused on the iPCG because it corresponds to the tongue and face regions, which are particularly important for language processing[102].

Principal component analysis (PCA) was applied to denoise the data by reducing dimensionality[103]. Specifically, voxel time series within each parcel were stacked for each trial, and principal components along with their explained variance were extracted. Consistent with prior research, subsequent analyses focused exclusively on the first three principal components corresponding approximately to 90% of variance.

To examine auditory-motor area coupling during the comprehension task, Gaussian-Copula Mutual Information (GCMI) was used to compute the mutual information between MEG time courses from two different brain areas. GCMI (and Weighted Pairwise Phase Consistency) tends to outperform other connectivity measures[104]. Furthermore, multivariate GCMI compared to univariate GCMI has been shown to be more advantageous for measures of speech to auditory and motor cortex coupling[103,105]. Specifically, GCMI was calculated between the left and right pSTG and the left and right IFG, iPCG, and SMA. Speech-motor area signals were systematically shifted to generate delays from 0 to 300 ms. Based on visual inspection, GCMI values were extracted at a 50 ms delay for each condition, as GCMI plateaued around 50 ms before declining at longer delays (Supplementary Fig. 3). To assess whether auditory-motor coupling exceeded chance levels and normalized GCMI values, true GCMI values were compared against the surrogate data. Surrogated data were generated by segmenting each trial into segments, shuffling them across iterations (N = 500), and estimating GCMI between the source time-series and the shuffled comparison time series. The segment length was kept constant across conditions, set at 80% of the cycle length corresponding to the syllabic rate frequency of each condition (e.g., 5 Syl/s = 160.0 ms, 11 Syl/s = 72.72 ms). This ensured that segment lengths varied across conditions, but relative length remained constant. The surrogate distribution was then used to generate z-transformed GCMI values by subtracting the true GCMI values from the mean of the surrogate distribution and dividing by its standard deviation. Normalization was performed within participants for each trial, frequency, and brain area parcel.

## Neural data statistics and reproducibility

For the statistical analysis of GCMI, generalized linear mixed models (GLMM) were employed (n = 57). No replication of the analysis was included. Specifically, the template model builder (TMB) function in the R statistical software package was used for model fitting, specifying the beta family to address heteroscedasticity issues. Prior to the analysis, the normalized GCMI data were scaled and min-max normalized to meet the beta distribution requirements, with data bounded between 0 and 1. Boundary values at 0 and 1 were adjusted by adding or subtracting a small value, calculated as ± (median/sample size). No multicollinearity issues were observed. The *ggplot2* package was employed for data visualization and plotting. All statistical analyses were corrected for multiple comparisons using the false discovery rate (FDR) method[106]. Post-hoc testing was conducted using estimated marginal means (EMM) from R to further evaluate significant findings, with FDR correction applied. For periodicity sensitivity comparisons, *emmtrends()* was used, while *emmeans()* was used for coupling analyses.

For the neural model, mutual information (GCMI) was analyzed as a function of syllabic rate, periodicity, ROI and hemisphere. The model incorporated interaction terms between syllabic rate, periodicity, and ROI and hemisphere and periodicity to assess how these factors jointly influence neural measures. Additionally, compression factor and sentence-level average word-frequency were included as covariates to control for their potential confounding effects. Random intercepts were included for both trials and participants to account for variability at each level, providing a robust framework to capture both within-subject (trial-level) and between-

subject variation. A third-degree polynomial (indicated as *variable³*) was selected based on the data distribution, its better model performance, and generalization capabilities. Note that ROI and hemisphere were treated as categorical variables, while all other variables were treated as metric variables. Given that no sum-to-zero coding scheme was used, fixed effects of ROI and hemisphere in the presence of interaction effects need to be considered cautiously and, therefore, are not interpreted.

Formular syntax for neural models

1. Big model:

$$Mutual\ information \sim syllabic\ rate^3 * periodicity * Region\ of\ interest$$
$$+ compression\ factor + sentence - level\ average\ word - frequency$$
$$+ hemisphere + (1|trials) + (1|subject)$$

2. Individual ROI model:

$$Mutual\ information \sim syllabic\ rate^3 * periodicity + compression\ factor$$
$$+ sentence - level\ average\ word - frequency + hemisphere$$
$$+ (1|trials) + (1|subject)$$

## Reporting summary
Further information on research design is available in the Nature Portfolio Reporting Summary linked to this article.

## Data availability
The anonymized preprocessed MEG and behavioral data are available on the Open Science Framework (OSF) as stated in Lubinus et al. [91]. Due to restrictions, the pseudonymized raw MRI and MEG data, as well as the unprocessed stimulus material, are not publicly available. The data required to reproduce the results are available on OSF[107].

## Code availability
All custom code central to the conclusions is available on OSF[107].

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

## Acknowledgements

We thank the Max Planck Institute for Empirical Aesthetics for funding this project. We thank Dr. Klaus Frieler for valuable advice on statistical analysis. AK is supported by the Medical Research Council (grant number MR/W02912X/1).

## Author contributions

S.K.: conceptualization, data curation, formal analysis, methodology, software, visualization, writing—original draft, review and editing; C.L.: data curation, investigation, methodology, project administration, software, writing—review and editing; C.A.K.: methodology, writing—review and editing; A.K.: methodology, visualization, writing—review and editing; J.M.R.: conceptualization, supervision, resources, methodology, writing—original draft, review and editing.

## Funding

## Competing interests

The authors declare no competing interests.
