## [Transparent Peer Review file · Communications Biology]

Effects of speech periodicity and speech rate on auditory-motor coupling during speech comprehension

Corresponding Author: Dr Johanna Rimmele

Version 0:

Reviewer comments:

Reviewer #1

(Remarks to the Author)

This paper examines the effect of rate and periodicity on speech comprehension and auditory-motor coupling at relevant frequencies in MEG data. The authors reason that the field of “oscillatory entrainment” – that often tests for such coupling – only rarely addresses the variation in rate and periodicity that natural speech contains. They report some interesting observations that suggest that less periodic speech is processed more readily, leading to better comprehension and increased coupling between, for example, inferior precentral gyrus and posterior superior temporal gyrus.

I enjoyed reading this manuscript which tackles a relevant and relatively unexplored question for the field. The dataset is complex and analysed in detail, yielding several important results. I do have some points that I hope can be clarified in a revised version of the manuscript. In short, some of the key claims made imply causality or directionality but do not always seem to be based on direct evidence.

The authors report that comprehension increases when speech is less regular. I wonder whether this might be because words that are more common and therefore easier to understand are spoken more irregularly. Conversely, difficult words could be spoken more regularly in an effort to make them easier to understand. In this scenario, the complexity of the word to be comprehended would determine regularity and not vice versa. Finally, words at the beginning of a phrase or sentence might be easier to understand because they are acoustically more isolated, but also be embedded in a more irregular context, for the same reason. Some additional analyses contrasting regular and irregular speech (e.g., position within a sentence/phrase, word length, complexity) might address this issue, and/or a more detailed discussion on how the authors believe the effect is produced.

Similarly, it remains unclear to me whether the observed auditory-motor coupling is functionally relevant, or a reflection of the fact that the regions coupled are driven by the same stimulus (but without a functional interaction). Maybe an analysis of directional connectivity between regions would answer this question? It might also be helpful to test whether the results observed are specific to auditory-motor coupling or whether “speech tracking” (i.e. “auditory-speech coupling”) shows a similar preference for certain rates and periodicity. If it does, it might have consequences for the interpretation of auditory-motor coupling, as a stronger response would be relayed from auditory to motor regions during the preferred stimulus (and might lead to stronger coupling only because this relayed response is stronger). If it does not, this result could strengthen the authors' results by demonstrating functional specificity for auditory-motor coupling.

The notion of neural oscillations possibly involved in the observed effects appears in several paragraphs, including those describing the study's rationale, but I wonder whether this is necessary, given that the authors' definition of speech tracking is independent of neural oscillations.

Could the authors please explain briefly why they used their specific measure of coupling, given that others exist (e.g., Gross et al 2021, NeuroImage)?

Reviewer #2

(Remarks to the Author)

The current manuscript reports an MEG study on speech perception, employing stimuli of different speech rates (by naturally varying speed of speaking and by compressing recordings). The authors find that syllabic rate modulates comprehension success and neural coupling between auditory and motor areas. The manuscript is succinct and well-written, presenting a clear set of findings.

I have some remarks that I would like the authors to address:

1. Comprehension accuracy

Comprehension success is currently modelled as the percentage of correctly produced words during recall of the sentence just heard at the beginning of a trial. It appears that modelling percentages with a Gaussian regression is not permissible because one assumption is that the dependent variable is unbounded, while percentages range from 0-100 (or 0-1 as proportions). Instead, it would be better to use a binomial model where a correctly recalled word counts as a success (coded as 1) and a word not recalled or recalled wrongly as a failure (coded as 0). The formula in lme4 could then be: "formula = cbind(number_of_successes,number_of_failures) ~ YOUR PREDICTORS, family = binomial"

One other problem that I see with the comprehension accuracy measure is also that sentences do have an internal grammatical and semantic/logical structure so that participants will also be able to guess some words even if they did not properly comprehend them. I.e., if they understood "Tisch", they can 100% predict that the corresponding article should be "der". This makes the accuracy of recalling the individual words dependent on each other. One solution would be to model comprehension success as follows: Instead of just entering one row per trial into the regression analysis, every single word becomes a row with an attached 1/0 binomial score. Then it would be possible to get surprisal values for each word as an estimate of how predictable it is given the context (the previous words) and use these as a (mean-centered) control predictor in the regression analyses; just as a main effect, no interactions are necessary. This results in the estimates for the manipulations of interest being adjusted for the predictability of words. However, it is important for this that all numerical predictors are mean-centered (actually standardised would be best so that they are all on the same scale) and that all categorical predictors are sum-coded or Helmert-coded. In addition, the model would need to know which words belong together into a sentence so that a sentence/trial ID needs to be added as a random effect. If I interpret the formula on lines 418-419 correctly, this is already the case.

However, this does then not yield a trial-level accuracy estimate but one for individual words. I am not sure whether this is what the authors had in mind. From the regression model, however, it may be possible to calculate the sentence-level accuracy based on the word-level estimates.

Finally, as a last point on this topic: Participants responses were manually transcribed (l. 403-404). How error-prone was the transcription process? Were there any measures in place to quantify transcription error (e.g., proof-reading by another coder or inter-rater reliability measurements etc.)?

2. Comprehension and auditory-motor coupling

It would be interesting to see how comprehension success relates to auditory-motor coupling since this is the underlying idea of the whole manuscript. Therefore, it would be good to see an analysis that, for example, predicts comprehension success from the neural coupling. (Or at least a good reason should be given why this is not possible.)

3. Experimental task

Could the auditory-motor coupling observed in this experiment also be driven by the task that participants had to perform? They knew that they listened to the sentences in order to then produce them themselves. Could this have amplified the engagement of the motor cortex already during listening because participants listened especially closely to the phonological forms or otherwise already shadowed the sentences while hearing them? It would be interesting to see whether similar effects are also observed if the task is only to answer a comprehension question or to type out the sentence (so that other motor areas than the language-articulator ones would be required to produce the answer).

4. Impact

One of the main impacts and take-home messages of the paper is somewhat hidden in the middle of the discussion and should be placed more prominently (also in the abstract). That is that the current findings suggest that it is very important to study natural speech and not just isochronous speech because different processes may be underlying these two (or rather that isochronous speech is convenient to study but not what the brain "cares" about in real-life).

Minor points:

* When discussing the role of theta, delta, and beta oscillations during speech comprehension, this study (<https://doi.org/10.1093/cercor/bhae479>) may also be mentioned because it fits well with the theme and shows that there are inter-dependencies between different oscillatory frequency bands and different levels of the phonological structure of speech.

* A table with the regression results for the comprehension LMER should be added (lines 115-127).

* At the end of the introduction, it would be nice to have 1-2 sentences foreshadowing the results to make it clear to the reader what the bigger picture is that is being contributed to.

* All regression formulas would benefit from some more explanation, for example, it is not clear what (1 | trial) refers to.

* I didn't quite understand the explanation around calculating MADs on lines 382-396. Could this be explained in more detail so that it is easier to follow, also for people who are not deeply into research on syllabic rate?

* In this paragraph, the numeric citations are missing.

* Line 438: The epochs are not clearly described. It should probably read -1000 ms relative to trial onset until 100 ms relative to trial offset.

* Line 441: "sensors with high z-values (>2) were rejected". What do the z-values refer to here?

Reviewer #3

(Remarks to the Author)

In this study authors set to uncover the effects of speech syllabic rate and periodicity (manipulating both independently) on theta coupling within the temporal and speech-motor areas. One of the key assumptions of this study is that temporal-motor speech areas theta phase coupling represents propagation of temporal predictions facilitated by the quasi-periodic structure of the syllabic information in speech. In this assumption coupling in theta should be maximal when comprehension is optimal, boosted by higher periodicity. Interestingly, authors find two opposite effects – with lower periodicity of syllabic information linked to stronger theta temporal-motor coupling and better comprehension; and simultaneously higher syllabic rate linked to worse comprehension and more theta phase coupling. This is a very informative finding that strongly contributes to the speech processing literature and challenges some of the current theoretical assumptions. While I believe that this paper is methodologically strong and conceptually sound, I have one major methods and one major interpretation concern that I would like authors to address in a revision.

Major

1. I do not believe that averaging between right and left hemisphere ROIs is justified. I understand the initial logic of similar theta coupling profiles between different hemispheres, but this does not imply that the relationships between theta temporal-motor coupling and the syllabic rate and periodicity will be the same across hemispheres. Therefore, in the revision I would suggest to either include hemisphere as a factor in the mixed model analysis (preferable) or conduct separate analyses on the two hemispheres. At the very least, these could be included as supplementary analyses, if effects in both hemispheres show the same profile.

2. My theoretical concern is a little more nuanced. It is a difficult argument to make – that stronger theta temporal-motor phase coupling simultaneously represents different and not fully related processes: (1) "optimal" processing (boosted by temporal predictions) for less periodic speech and (2) "processing effort" for faster speech. In the first instance, it is not made clear why speech with less periodic syllabic rate will result in stronger temporal predictions, and what such predictions can be based on (higher levels of linguistic hierarchy?). This needs to be further clarified in the discussion. Second, an alternative explanation could be made – that theta temporal-motor coupling increases when temporal structure of the speech is weakened or strained (faster speech syllabic rate and less periodicity), and reliance on temporal syllable-level temporal predictions is challenged. However, comprehension doesn't suffer in naturalistic speech (less periodic) when other linguistic levels of information are not necessarily affected (e.g. speakers who talk faster would still preserve natural intonation, indicative of phasal boundaries etc.) and temporal prediction based on syllabic boundaries may play a limited role.

Minor

1. In the introduction lines 90-97 a lot of concepts are introduced in quick succession; more unpacking of the literature can be done – especially how phase coupling in different bands is linked to temporal prediction (and at what level).

2. At the moment, the key manipulation of speech rate versus periodicity of stimuli is not explained very clearly in the introduction and only after reading Methods section the narrative becomes clear.

3. Given the literature the authors review on the role of delta-theta coupling in temporal predictions, it is not clear why authors did not conduct such analyses themselves? Especially, given their claim that more naturalistic and less periodic stimuli could produce greater temporal predictions that are not linked exclusively to syllabic level information.

Version 1:

Reviewer comments:

Reviewer #1

(Remarks to the Author)

Thank you for this thoroughly revised manuscript. My concerns have been addressed.

Reviewer #2

(Remarks to the Author)

I thank the authors for their thoughtful consideration of my and the other reviewers' comments. I am satisfied with the responses to my comments.

In particular, I agree that the beta models are a better strategy than binomial regression models.

My remaining comments are:

Since the "word complexity" variable consists only of the average word frequency in the sentence, it should also be called something with frequency. "Complexity" in the context of sentences would indicate that other features (such as the number of

nodes or clauses) plays into the measure, too.

The transcription error would be relatively easy to quantify and I encourage the authors to add this, as it is good scientific practice. A quick way to do this would have another person check a random sample of 10% of the trials, listen to the audio recordings and compare the transcript. This can then be the basis of calculating inter-rater agreement.

The pattern of results for the analysis of whether speech comprehension is predicted by auditory-motor coupling are quite complex. I can't find the information how the categorical predictors (syllable rate, periodicity) were coded. Only if a sum-to-zero coding scheme was used (e.g., sum coding) is it possible to interpret main effects in the presence of interaction effects. This information should be added to the manuscript (also for the other models).

Why is the corresponding figure only in the supplemental material? I would find it nicer, also for the reader, to have this in the main text.

Regarding the amplification of motor area engagement by the sentence repetition task, I find the argument that a previous study by the authors did also find similar effects with a different task that didn't involve speaking. This should find its way into the manuscript and not just in the response to the reviewers. This would also not let the limitation hang like it is now.

In the regression tables in the supplemental material, one of the columns is called "Statistics". What does it mean?

Is it really necessary to correct the p values with FDR? This is an honest question. Within the same model, one gets multiple p values, but they come from the same test, so to say, because they derive from the solution to the modelling process to which all predictors contributed jointly and at the same time. Across the different analyses reported in the experiment, different aspects of the collected data were analysed, so I am also not so sure about the necessity of the FDR correction. It definitely won't hurt, though.

In the caption of each regression results table, please specify what model this was (e.g., beta model) and ideally add the formula.

Did all models converge without issues? I can see that the random effects often have very small values (close to 0). This can be an indication of convergence issues.

Please add the tables as actual tables, not as screenshots.

The readers would also profit from more human-readable predictor labels. E.g., $\text{poly}(\text{Syllabic rate}, 3)^2 \cdot \text{MAD}$ could easily be written as "Syllabic rate² x MAD" (with the ² meaning a superscript).

Reviewer #3

(Remarks to the Author)

Version 2:

Reviewer comments:

Reviewer #2

(Remarks to the Author)

I thank the authors for also thoughtfully addressing my comments in this round and I am now fully satisfied with the responses. I congratulate the authors on this nice piece of work!

Dear Professor Blank, dear reviewers,

Many thanks for the positive evaluation and the constructive feedback. We performed an extensive revision following the recommendations, which helped us to improve the manuscript.

In summary, we revised our statistical analysis now using GLMMs to account for issues due to the bounded distribution. We included hemisphere as predictor of the neural coupling and of the periodicity effects on neural coupling. An estimate for sentence level word complexity was included in the behavioral and neural GLMM analyses. We performed two types of directed connectivity measures to clarify whether the speech auditory-motor coupling was related to a common cause or reflected top-down vs. bottom-up processing. To provide further insides into the relationship between neural coupling and speech comprehension, we added a GLMM predicting speech comprehension behavior based on the neural coupling, as well as moderations of periodicity and syllabic rate. We revised the discussion of the comprehension-coupling relationship accordingly. We largely revised the introduction to clarify our motivation for investigating theta-theta coupling, as well as the background with regard to the neural oscillatory account and the impact of our research.

Below, we respond to the editor's comments, they are in detail responded to in response to the reviewer comments in the letter. All issues raised by the reviewer are responded to below.

Editorial comments:

(1) whether speech regularity is driven by word complexity

Following the reviewers' suggestions we now added a measure of word complexity as control variable to our statistical analysis (R1 issue 2, R2 issue 2). Given that our analysis was committed to the sentence level, we computed the average word frequency per sentence as estimate for sentence-wise word complexity. Word complexity did not have an effect on speech comprehension, however, higher word complexity was related to increased coupling. The periodicity effects on comprehension and neural coupling were still observed after the control variable has been added.

(2) functional relevance of auditory-motor coupling through connectivity analysis and comparisons with auditory-speech coupling.

We performed two measures of directed connectivity, transfer entropy applying whitening (TE) and delays of Gaussian Copula Mutual Information (GCMI) (see R1 issue 3). As the findings need to be interpreted cautiously, we only included them here for the reviewers' inspection. However, if recommended we would be happy to include them to the supplement. The findings suggest that both bottom-up and top-down effects contribute to the coupling. The manuscript has been revised accordingly.

(3) justify or revise the approach to averaging hemispheric ROIs by including hemisphere as a factor or conducting separate analyses

Hemisphere has now been included in the statistical analysis of neural coupling as fixed effect and its interaction with periodicity (R3 Issue 3). Although, overall coupling was stronger in the left hemisphere, periodicity effects did not vary across hemispheres.

further clarify how (4) theta temporal-motor coupling simultaneously supports both optimal processing and processing effort in different contexts.

We now directly analyzed the relation between comprehension and neural coupling, and the statistics have been added to the results section (a figure has been added to the supplement). The findings are discussed in the discussion section (R2 issue 3 and R3 issue 2). Based on these findings, we now provide

a "single mechanism" explanation and emphasize that further research is necessary to confirm such speculation.

Reviewers' comments:

Reviewer #1 (Remarks to the Author):

This paper examines the effect of rate and periodicity on speech comprehension and auditory-motor coupling at relevant frequencies in MEG data. The authors reason that the field of "oscillatory entrainment" – that often tests for such coupling – only rarely addresses the variation in rate and periodicity that natural speech contains. They report some interesting observations that suggest that less periodic speech is processed more readily, leading to better comprehension and increased coupling between, for example, inferior precentral gyrus and posterior superior temporal gyrus.

R1 Issue 1:

I enjoyed reading this manuscript which tackles a relevant and relatively unexplored question for the field. The dataset is complex and analysed in detail, yielding several important results. I do have some points that I hope can be clarified in a revised version of the manuscript. In short, some of the key claims made imply causality or directionality but do not always seem to be based on direct evidence.

Thank you for the positive evaluation and helpful feedback.

We have revised the manuscript to remove wording that may have implicated causality, as we agree that given the correlational nature of our study no claims of causality are possible.

e.g. We revised the title to: "Effects of speech periodicity and speech rate on auditory-motor coupling during speech comprehension "

Additionally, we responded to all the issues below.

R1 Issue 2:

The authors report that comprehension increases when speech is less regular. I wonder whether this might be because words that are more common and therefore easier to understand are spoken more irregularly. Conversely, difficult words could be spoken more regularly in an effort to make them easier to understand. In this scenario, the complexity of the word to be comprehended would determine regularity and not vice versa. Finally, words at the beginning of a phrase or sentence might be easier to understand because they are acoustically more isolated, but also be embedded in a more irregular context, for the same reason. Some additional analyses contrasting regular and irregular speech (e.g., position within a sentence/phrase, word length, complexity) might address this issue, and/or a more detailed discussion on how the authors believe the effect is produced.

We had discussed a related point as a possible explanation for the effects of periodicity in the previous manuscript version. Ten Over & Martin (2021) (and others) had suggested that word frequency and word duration are related. They hypothesized that this way linguistic top-down predictions affect not only the periodicity of the speech signal by impacting the duration between words but also the ability of the brain to process aperiodic signals related to an oscillatory mechanism. Our findings that lower periodicity is related to higher comprehension, may reflect (1) an "auditory sweet spot", with neuron populations in auditory cortex being tuned to respond to speech with a lower degree of periodicity (Pittman-Poletta et al., 2021), and (2) temporal predictions related to higher-level linguistic processes

and/or prosody-related processes (Ten Over & Martin, 2021). We revised the introduction and discussion to clarify this.

Following the reviewers and editors' recommendation we now added a control variable for word complexity. Average word frequency was computed per sentence (as estimate of sentence level word complexity) and added as predictor to the GLMMs for comprehension and neural coupling. Word frequency was computed at the sentence level, because both our neural measure (GCMI) and our periodicity measure are performed at the sentence level. Periodicity cannot be directly or reasonably be measured at the word/syllable level. Word frequency was computed using the Wortschatz Corpus (Mixed-Typical 2011 dataset). Our new results show that word frequency did not predict speech comprehension, but was positively associated with neural coupling (see below). Importantly, periodicity effects were still observed when the control variable was added to the GLMMs. The methods, results and discussion sections have been revised accordingly.

In the introduction:

I. 57: "Aubanel and Schwartz¹⁶ concluded that predictive cues related to bottom-up isochrony and top-down naturalness may be combined to aid speech comprehension. "

I. 73: "Thereby, predictions may vary in strength for periodic and non-periodic speech, as features, such as prosody or higher-level linguistic processing affect the temporal predictability of speech^{7,11,12}."

I. 312: "We tested, furthermore, whether the periodicity effects on comprehension were reflecting effects of word complexity. When speech is produced, word complexity may impact the periodicity of the speech signal^{7,16}, with more frequent words being preceded by shorter words⁷. In our study, however, the sentence level average word frequency did not predict speech comprehension, while it did affect auditory-motor coupling. Importantly, periodicity effects were observed when word frequency was controlled for."

Method section, I. 486: "Sentence-wise speech comprehension performance (measured as % correct) was modeled as a function of syllabic rate, periodicity, compression factor, syllable number, stimulation order, and word complexity. The word complexity was estimated based on the average word frequency data per sentence from the Leipziger Wortschatz Corpus (Mixed-Typical 2011 dataset)."

R1 Issue 3

Similarly, it remains unclear to me whether the observed auditory-motor coupling is functionally relevant, or a reflection of the fact that the regions coupled are driven by the same stimulus (but without a functional interaction). Maybe an analysis of directional connectivity between regions would answer this question? It might also be helpful to test whether the results observed are specific to auditory-motor coupling or whether "speech tracking" (i.e. "auditory-speech coupling") shows a similar preference for certain rates and periodicity. If it does, it might have consequences for the interpretation of auditory-motor coupling, as a stronger response would be relayed from auditory to motor regions during the preferred stimulus (and might lead to stronger coupling only because this relayed response is stronger). If it does not, this result could strengthen the authors' results by demonstrating functional specificity for auditory-motor coupling.

To analyze directional connectivity, we used Transfer Entropy (TE), as implemented in the TRENTOOL toolbox (Lindner, Vicente, Priesemann, Wibral, 2011). As TE is a problematic measure when using narrow band (band-passed) data (Daube, Gross, Ince, 2022), we applied a novel approach with whitening (Daube, Gross, Ince, 2022). We furthermore compared the TE analysis to a GCMI delay analysis. Our findings suggest one ROI pair (left iPCG-pSTG) with zero delay (measured consistent

across methods). In this case, a common cause (i.e. auditory and motor areas may be driven by the same stimulus) cannot be excluded. For all other areas, we either observed a positive (motor leading) or negative (auditory leading) delay. This suggests top-down and bottom-up effects, likely not caused by the stimulus as a common cause. However, the directional connectivity analysis is limited: In general, this method must be interpreted cautiously as lags and even directionality may depend on analysis choices. Furthermore, given the short stimulus length of our trials, the lack of clear peaks in many of the ROI-pairs, and some inconsistencies across the two analysis methods, our findings need to be interpreted cautiously. Therefore, we only included them here for the reviewers' inspection. Of course, if suggested by the reviewer, we can include them in the supplement.

The findings of the analyses are pasted below for the reviewers' inspection:

TE was estimated using the TRENTTOOL toolbox (version 3.0) in MATLAB. Before estimating TE, the data were whitened and then processed with a Hilbert transform to obtain the signal in the narrow theta band. As TRENTTOOL computes TE for non-negative delays, analyses for negative delays were performed by swapping the source and target signals and applying the same procedure.

R1 Issue 4

The notion of neural oscillations possibly involved in the observed effects appears in several paragraphs, including those describing the study's rationale, but I wonder whether this is necessary, given that the authors' definition of speech tracking is independent of neural oscillations.

Our research is motivated by neural oscillatory models of speech segmentation and auditory-motor interaction. As outlined in the discussion, neural oscillatory models assume a special role for periodicity. Therefore, it seems relevant to discuss the notion of neural oscillations. While we discuss neural oscillatory accounts and their theoretical relevance for our study, we chose to be cautious when reporting research that has not directly tested whether oscillations are involved by using the term "speech tracking" or "neural coupling". In our study, we investigate auditory-motor coupling in the phase-phase space, however, we do not directly test the involvement of neural oscillations. We restructured and revised the introduction and revised the discussion to clarify these points.

I. 63: "An influential neural oscillatory account emphasizes how cortical oscillations operate upon syllabic-level periodicity to facilitate speech comprehension^{17–21}. According to this, populations of neural theta-band (4–8 Hz) oscillations in auditory cortex align their high excitability phases^{21,22} to the slow temporal fluctuations in the acoustic speech envelope at the syllabic scale to aid syllabic segmentation, a basis for comprehension^{6,19,23–27}. Because the involvement of neural oscillations is typically not directly shown, we use the descriptive term *speech tracking* when reporting research on the alignment of neural activity to the speech acoustics. The neural oscillatory account assumes stimulus periodicity to be a relevant feature, with higher degrees of periodicity being expected to increase the entrainment strength and subsequently comprehension^{28,29}. Additionally, to such rhythm processing in auditory cortex, predictions from the motor cortex may be particularly relevant for processing the complex temporal structure of speech^{30–33}. Thereby, predictions may vary in strength for periodic and non-periodic speech, as features, such as prosody or higher-level linguistic processing affect the temporal predictability of speech^{7,11,12}. Interestingly, temporal predictions generated in speech-motor areas about upcoming sensory events during speech perception have also been linked to neural oscillations^{30,34–36}. [...]"

R1 Issue 5

Could the authors please explain briefly why they used their specific measure of coupling, given that others exist (e.g., Gross et al 2021, NeuroImage)?

Gross et al. (2021) suggest that "Weighted Pairwise Phase Consistency (WPPC) and GCMI tend to outperform other connectivity measures". We chose multivariate GCMI as measure as it allows us to base the analysis on multiple ICA components extracted from a ROI and thus optimized the signal to noise ratio (see also: De Clercq et al., 2023). Furthermore, another study suggests that multivariate GCMI is more advantageous compared to univariate GCMI for looking at speech to auditory and motor cortex coupling (Chalas et al., 2022; De Clercq et al., 2023). Multivariate GCMI has not been considered in Gross et al. (2021), however, in combination these publications suggest that multivariate GCMI is a very suitable approach for our study.

We added a sentence to justify the methods choice in the manuscript,

I. 557: " GCMI (and Weighted Pairwise Phase Consistency) tends to outperform other connectivity measures¹⁰³. Furthermore, multivariate GCMI compared to univariate GCMI has been shown to be more advantageous for measures of speech to auditory and motor cortex coupling^{102,104}."

Reviewer #2 (Remarks to the Author):

The current manuscript reports an MEG study on speech perception, employing stimuli of different speech rates (by naturally varying speed of speaking and by compressing recordings). The authors find that syllabic rate modulates comprehension success and neural coupling between auditory and motor areas. The manuscript is succinct and well-written, presenting a clear set of findings.

I have some remarks that I would like the authors to address:

Thank you for the positive evaluation and helpful comments. All comments are responded to below.

R2- Issue 1:

1. Comprehension accuracy

Comprehension success is currently modelled as the percentage of correctly produced words during recall of the sentence just heard at the beginning of a trial. It appears that modelling percentages with a Gaussian regression is not permissible because one assumption is that the dependent variable is unbounded, while percentages range from 0-100 (or 0-1 as proportions). Instead, it would be better to use a binomial model where a correctly recalled word counts as a success (coded as 1) and a word not recalled or recalled wrongly as a failure (coded as 0). The formula in lme4 could then be: "formula = cbind(number_of_successes,number_of_failures) ~ YOUR PREDICTORS, family = binomial"

Thank you for pointing this out. We agree that our previous approach did not take into account the bounded nature of our comprehension measure. However, the suggested binomial model is not suitable for our analysis, as the periodicity measure and the neural coupling measure require the analysis of sentence-wise data, not word-wise data. We now revised all our analyses (behavioral and neural models) in order to deal with the distribution issues the reviewer pointed out. We revised all analyses to now use a generalized linear mixed model (glmmTMB) with a beta distribution with our sentence-wise data. This allows to model data bounded between 0-1. We corrected for values at 0 and 1 by adding/subtracting small values (formula: \$V \pm (\text{median}/\text{sample size})\$ ).

Our results did not change (other than numerically).

The manuscript was revised in the methods and results section accordingly.

Methods section:

I. 481: "The behavioral data (n=57) were analyzed using generalized linear mixed models (GLMM). Model fitting was performed using the Template Model Builder (TMB) in R, specifying the beta family. All analyses used the glmmTMB package with a beta distribution to appropriately model sentence-wise data bounded between 0 and 1. To address boundary values at 0 and 1, we adjusted the data by adding or subtracting a small value, calculated as \$\pm (\text{median}/\text{sample size})\$. No multicollinearity issues were observed. [...]"

I. 573: "For the statistical analysis of GCMI, generalized linear mixed models (GLMM) were employed (n=57). Specifically, the template model builder (TMB) function in the R statistical software package was used for model fitting specifying the beta family, [...]"

The results section has been revised accordingly.

R2 Issue 2

One other problem that I see with the comprehension accuracy measure is also that sentences do have an internal grammatical and semantic/logical structure so that participants will also be able to

guess some words even if they did not properly comprehend them. I.e., if they understood "Tisch", they can 100% predict that the corresponding article should be "der". This makes the accuracy of recalling the individual words dependent on each other. One solution would be to model comprehension success as follows: Instead of just entering one row per trial into the regression analysis, every single word becomes a row with an attached 1/0 binomial score. Then it would be possible to get surprisal values for each word as an estimate of how predictable it is given the context (the previous words) and use these as a (mean-centered) control predictor in the regression analyses; just as a main effect, no interactions are necessary. This results in the estimates for the manipulations of interest being adjusted for the predictability of words. However, it is important for this that all numerical predictors are mean-centered (actually standardised would be best so that they are all on the same scale) and that all categorical predictors are sum-coded or Helmert-coded. In addition, the model would need to know which words belong together into a sentence so that a sentence/trial ID needs to be added as a random effect. If I interpret the formula on lines 418-419 correctly, this is already the case. However, this does then not yield a trial-level accuracy estimate but one for individual words. I am not sure whether this is what the authors had in mind. From the regression model, however, it may be possible to calculate the sentence-level accuracy based on the word-level estimates.

As the Reviewer points out, for our purposes a sentence level measure of comprehension accuracy is wanted. The suggested analysis is interesting, but unfortunately beyond the scope of our manuscript. Note that we added a sentence level measure of word complexity (average word frequency per sentence) as a control variable. (see also R1-Issue 2 and response to the editorial comments)

R2 Issue 3

Finally, as a last point on this topic: Participants responses were manually transcribed (l. 403-404). How error-prone was the transcription process? Were there any measures in place to quantify transcription error (e.g., proof-reading by another coder or inter-rater reliability measurements etc.)?

The responses were manually described by an experienced native-German researcher (the co-first author Dr Christina Lubinus), and the transcriptions were checked periodically. No measures regarding transcription error were taken. We have now added this information to the manuscript methods section.

L. 474: " In the context of Lubinus et al.⁹¹, the recorded participants' responses were manually transcribed by the co-first author to produce text excerpts (note that the transcription error was not quantified)."

R2- Issue 3:

2. Comprehension and auditory-motor coupling

It would be interesting to see how comprehension success relates to auditory-motor coupling since this is the underlying idea of the whole manuscript. Therefore, it would be good to see an analysis that, for example, predicts comprehension success from the neural coupling. (Or at least a good reason should be given why this is not possible.)

We now added the suggested analysis to the results section, discussed the findings in the discussion section and added a figure in the supplement (also pasted below). In the previous version of the manuscript, we had described the relation between behavior and auditory-motor coupling based on separate analyses. Higher speech comprehension was predicted by lower periodicity (higher MAD) and lower syllabic rates. Higher speech coupling was predicted by lower periodicity (higher MAD) and higher syllabic rate. This led us to speculate about a double dissociation of behavior and coupling.

As suggested by the reviewer (see also R3 issue 2), we now perform a direct analysis of whether speech comprehension is predicted by auditory-motor coupling. In a GLMM (Supplementary Table 3) we predict comprehension by the auditory-motor coupling (GCMI), the periodicity, the syllabic rate, as well as the interactions between coupling and periodicity and coupling and syllabic rate. The interactions test whether the relationship between comprehension and coupling (i.e. the slope) is moderated by the syllabic rate and/or periodicity. We find a fixed effect of coupling on comprehension, with lower coupling being related to higher comprehension. Periodicity and syllabic rate both show a fixed effect after adjusting for the effect of coupling, with lower periodicity (higher MAD) and lower syllabic rates related to lower comprehension. Additionally, we find a moderating effect of the syllabic rate on the effect of coupling on comprehension (two-way interaction), where periodicity moderated the effects of coupling on comprehension (two-way interaction). This was further specified by a 3-way interaction, suggesting that the relationship between comprehension and coupling was jointly moderated by syllabic rate and periodicity. Thereby, periodicity seems to particularly impact the moderating effect of syllabic rate on the comprehension-coupling relationship at slow syllabic rates (5 syl/sec, 11 syl/sec), with higher coupling related to higher comprehension for the high periodicity speech, whereas for the low periodicity speech no strong comprehension-coupling relationship seems present at slow syllabic rates. At higher syllabic rates for all periodicity conditions, low coupling was related to high comprehension.

We revised the manuscript accordingly and discussed possible interpretations in the discussion section:

Abstract, l. 41: " Comprehension improved with lower periodicity and declined at faster rates, with the syllabic rate and periodicity moderating the coupling-comprehension relationship."

Introduction:

l. 117: "Auditory-motor coupling predicted comprehension in a complex relationship that was moderated by the syllabic rate and periodicity of speech. The findings are in line with a sweet spot for speech processing of natural, less periodic speech."

Results:

l. 261: "In a direct analysis (GLMM) of the relationship of speech comprehension and auditory-motor coupling, we predict comprehension by the auditory-motor coupling (GCMI), the periodicity, the syllabic rate, as well as the interactions between coupling periodicity and syllabic rate. The interactions tests whether the relationship between comprehension and coupling (i.e. the slope) is moderated by the syllabic rate or periodicity. A fixed effect of GCMI ($b = 3.46$, $SE = 0.15$, $p < 0.001$), [...]".

Discussion:

l. 288: "We observed a complex relationship between auditory-motor coupling and comprehension that was moderated by the syllabic rate and periodicity of speech. These results possibly reflect both: increased temporal predictions from the motor system to facilitate comprehension in demanding listening situations and optimal auditory cortex responsiveness to natural low periodicity speech."

l. 349: "Interestingly, a direct analysis revealed a complex relationship between speech comprehension performance and auditory-motor coupling that was jointly moderated by the syllabic rate and the speech periodicity (Supplementary Figure 3). A fixed effect suggests that lower comprehension was related to higher auditory-motor coupling. This relationship, however, was further specified by two-way and three-way interactions. Both high and low auditory-motor coupling could predict high comprehension dependent on the periodicity and syllabic rate context. We speculate that the findings can be accounted for by a complex interplay of lower-level temporal processing in auditory cortex and higher-level temporal prediction from the speech motor cortices, possibly related to linguistic and

prosodic information³³. More specifically, at slower syllabic rates that correspond to our natural speech production rates, the motor cortex may provide temporal predictions of the speech signal. In the case of high periodicity speech such temporal motor predictions may be useful, as the periodic signal lacks the natural temporal structure of speech. This possibly resulted in higher auditory-motor coupling being related to higher comprehension when syllabic rates are slow and periodicity is high. In the case of low periodicity speech presented at slow syllabic rates, auditory cortex processing may be optimal -given that the natural temporal structure of speech is preserved- and individuals may more strongly rely on sensory representations. For fast syllabic rates given the demanding listening situation, temporal predictions from the motor cortex may be expected to be particularly useful, as shown for simple tone sequences⁸⁸. Others, however, did not find increased auditory-motor engagement at fast speech rates⁸⁵. We find that for fast speech rates higher comprehension was related to lower coupling. A possibility is that because of our experience in speech production the speech motor cortex provides exact temporal prediction for speech spoken at a natural slow tempo. At very fast rates outside of our production abilities, in contrast, predictions may be detrimental. Three-way interactions, however, should be interpreted cautiously and further research should explore this complex comprehension-coupling relationship. Furthermore, while the engagement of the motor system has been shown even during passive listening to speech^{38,39}, and with different tasks⁴¹, it is possible that our speech comprehension task emphasized the auditory-motor coupling, due to the requirement of verbally repeating the sentence.

In summary, our findings suggest a sweet spot with highest comprehension for low periodicity speech. This likely involves both processing within auditory cortex, and auditory-motor coupling facilitating comprehension in temporally impoverished listening situations (high periodicity speech)."

I. 393: "with a complex relationship between comprehension and coupling."

Supplementary Figure 3. Speech periodicity and syllabic rate moderate the coupling-comprehension relationship. The figure displays how the three-way interaction between auditory-motor coupling (GCMI), syllabic rates and periodicity levels predicts speech comprehension. Lines within each panel represent quartiles of MAD, ranging from low MAD (high periodicity) to high MAD (low periodicity). At slower syllabic rates (5 syllables/s, 11 syllables/sec) for higher periodicity (low MAD) speech, higher auditory-motor coupling (GCMI)

was related to higher comprehension. Less of a relationship seems evident for low periodicity speech (at 5 syllables/s). At higher syllabic rates higher comprehension was related to lower GCMI.

R2- Issue 4:

3. Experimental task

Could the auditory-motor coupling observed in this experiment also be driven by the task that participants had to perform? They knew that they listened to the sentences in order to then produce them themselves. Could this have amplified the engagement of the motor cortex already during listening because participants listened especially closely to the phonological forms or otherwise already shadowed the sentences while hearing them? It would be interesting to see whether similar effects are also observed if the task is only to answer a comprehension question or to type out the sentence (so that other motor areas than the language-articulator ones would be required to produce the answer).

The sentence repetition task has been typically (among some other tasks) used to test speech intelligibility (Beukelman & Yorkestone, 1979). In our previous behavioral study (Lubinus et al., 2023) we found motor related processes to correlate with speech comprehension in two different types of tasks: the sentence repetition task used here and a task where no speech production was required (word order judgement). This suggests that the motor system engagement is not merely due to the type of task. This is also in line with previous research suggesting a motor system engagement during passive listening to speech (.e.g., Park et al., 2015; Assaneo & Poeppel, 2018). However, we agree that it is possible that our task emphasized the motor system engagement. However, we observed differential effects which were in the focus of our interest.

We now discuss this in the discussion section.

I. 371: “Furthermore, while the engagement of the motor system has been shown even during passive listening to speech^{38,39}, and with different tasks⁴¹, it is possible that our speech comprehension task emphasized the auditory-motor coupling, due to the requirement of verbally repeating the sentence. In summary, our findings suggest a sweet spot with highest comprehension for low periodicity speech.”

R2- Issue 5:

4. Impact

One of the main impacts and take-home messages of the paper is somewhat hidden in the middle of the discussion and should be placed more prominently (also in the abstract). That is that the current findings suggest that it is very important to study natural speech and not just isochronous speech because different processes may be underlying these two (or rather that isochronous speech is convenient to study but not what the brain "cares" about in real-life).

We agree and have revised the abstract and the discussion to emphasize this:

I. 35: “Using magnetoencephalography, we investigated how natural variation in syllabic-level periodicity affects comprehension and auditory-motor coupling [...].

L 43: “These findings suggest a sweet spot for natural, less periodic speech rhythms that support optimal processing and emphasize the necessity to investigate natural speech.”

Minor

points:

* When discussing the role of theta, delta, and beta oscillations during speech comprehension, this study (<https://doi.org/10.1093/cercor/bhae479>) may also be mentioned because it fits well with the theme and shows that there are inter-dependencies between different oscillatory frequency bands and different levels of the phonological structure of speech.

We added the reference in l. 67.

* A table with the regression results for the comprehension LMER should be added (lines 115-127).

Given the several analyses we computed, we report all the relevant statistics in the text and added a complete table in the supplement. This has now been made clearer.

l. 128: "For the comprehension model, a generalized linear mixed model (GLMM; Full Statistics in Supplementary Table 1) [...]"

* At the end of the introduction, it would be nice to have 1-2 sentences foreshadowing the results to make it clear to the reader what the bigger picture is that is being contributed to.

We had a short summary in the previous version of the manuscript, however, extended it now.

l. 114: "Our findings suggest that natural, less periodic speech rhythms are optimally processed, with lower periodicity being associated with stronger auditory-motor coupling and with higher comprehension. Particularly, coupling between the pSTG and iPCG, but also pSTG and IFG and SMA, was sensitive to periodicity. Auditory-motor coupling predicted comprehension in a complex relationship that was moderated by the syllabic rate and periodicity of speech. The findings are in line with a sweet spot for speech processing of natural, less periodic speech."

* All regression formulas would benefit from some more explanation, for example, it is not clear what (1 | trial) refers to.

Our analysis is detailed in the methods, so that it can be reproduced. We additionally provided the syntax of the mixed model ("formula syntax"), as commonly used in R or Matlab, and as is customary done. It is beyond the scope of the manuscript to explain the syntax further, but researchers will be able to exactly reproduce our models based on the provided syntax. We now pointed out more clearly that this is the R code corresponding to the details in the methods. Additionally, our code will be posted at a repository upon acceptance.

Headers have been changed from "equation" to "Formula syntax"

* I didn't quite understand the explanation around calculating MADs on lines 382-396. Could this be explained in more detail so that it is easier to follow, also for people who are not deeply into research on syllabic rate?

We revised this section to improve clarity:

l. 446: "To measure periodicity, we used R (version 4.3.3) to calculate the median absolute deviation (MAD) of the inter-syllabic-nuclei intervals for each sentence. High periodicity corresponds to small variation in the inter-syllabic-nuclei intervals (i.e., small MAD values). The method has been previously used to quantify periodicity, as it is relatively robust to outliers^{95,96}. The inter-syllabic-nuclei intervals were calculated the following way: Using Praat and a syllable nuclei detection script⁹⁷, we extracted the timing of each syllable nucleus from each sentence recording. For each sentence, the MAD of the timing of all syllable nuclei was computed, resulting in one MAD value per sentence [...]"

* In this paragraph, the numeric citations are missing.

This has been fixed.

* Line 438: The epochs are not clearly described. It should probably read -1000 ms relative to trial onset until 100 ms relative to trial offset.

This has been fixed.

l. 515: "The data were down-sampled to 500 Hz and epoched (-1000ms relative to trial onset until +100ms relative to trial offset), resulting in epochs of varying lengths."

* Line 441: "sensors with high z-values (>2) were rejected". What do the z-values refer to here?

This has been revised.

l. 518: "with high z-values computed across sensors ($z > 2$) were rejected. Finally, ICA uses the infomax algorithm⁹⁸ to correct eye blink, eye movement, and heartbeat artifacts."

Reviewer #3 (Remarks to the Author):

In this study authors set to uncover the effects of speech syllabic rate and periodicity (manipulating both independently) on theta coupling within the temporal and speech-motor areas. One of the key assumptions of this study is that temporal-motor speech areas theta phase coupling represents propagation of temporal predictions facilitated by the quasi-periodic structure of the syllabic information in speech. In this assumption coupling in theta should be maximal when comprehension is optimal, boosted by higher periodicity. Interestingly, authors find two opposite effects – with lower periodicity of syllabic information linked to stronger theta temporal-motor coupling and better comprehension; and simultaneously higher syllabic rate linked to worse comprehension and more theta phase coupling. This is a very informative finding that strongly contributes to the speech processing literature and challenges some of the current theoretical assumptions. While I believe that this paper is methodologically strong and conceptually sound, I have one major methods and one major interpretation concern that I would like authors to address in a revision.

Major

R3- Issue 1:

1. I do not believe that averaging between right and left hemisphere ROIs is justified. I understand the initial logic of similar theta coupling profiles between different hemispheres, but this does not imply that the relationships between theta temporal-motor coupling and the syllabic rate and periodicity will be the same across hemispheres. Therefore, in the revision I would suggest to either include hemisphere as a factor in the mixed model analysis (preferable) or conduct separate analyses on the two hemispheres. At the very least, these could be included as supplementary analyses, if effects in both hemispheres show the same profile.

We now included hemisphere as factor in the neural model. We observed a fixed effect of hemisphere, suggesting overall stronger coupling in the left hemisphere. However, no interaction of the factors hemisphere and periodicity were observed, suggesting similar effects of periodicity on the coupling in both hemispheres. Furthermore, the results otherwise did not change with the inclusion of hemisphere. We revised the results (l. 163, l. 169, l. 170) and methods section (l. 579 ff.) to include those findings (Full statistics in supplementary Table 2).

R3 Issue 2

2. My theoretical concern is a little more nuanced. It is a difficult argument to make – that stronger theta temporal-motor phase coupling simultaneously represents different and not fully related processes: (1) “optimal” processing (boosted by temporal predictions) for less periodic speech and (2) “processing effort” for faster speech. In the first instance, it is not made clear **why speech with less periodic syllabic rate will result in stronger temporal predictions**, and what such predictions can be based on (higher levels of linguistic hierarchy?). This needs to be further clarified in the discussion. Second, an alternative explanation could be made – **that theta temporal-motor coupling increases when temporal structure of the speech is weakened or strained** (faster speech syllabic rate and less periodicity), and reliance on temporal syllable-level temporal predictions is challenged. However, comprehension doesn’t suffer in naturalistic speech (less periodic) when other linguistic levels of information are not necessarily affected (e.g. speakers who talk faster would still preserve natural intonation, indicative of phasal boundaries etc.) and temporal prediction based on syllabic boundaries may play a limited role.

MEG recordings tap into neural activity stemming from larger neuron populations. Therefore, it is likely that multiple processes in parallel may be reflected in the measured coupling signal. The theta-band also has been implicated in various processes. A single mechanism explanation, however, would be more parsimonious, and we agree with the reviewer that it would be preferable.

Importantly, we have now performed a direct analysis of whether speech comprehension is predicted by auditory-motor coupling (see also editorial comment and R2 issue 3). In a glmmTMB we predict comprehension by the auditory-motor coupling (GCM), the periodicity, the syllabic rate, as well as, the interactions between coupling and periodicity and coupling and syllabic rate. The interactions tests whether the relationship between comprehension and coupling (i.e. the slope) is moderated by the syllabic rate and/or periodicity. This analysis draws a more complex picture than the separate statistic models for comprehension and coupling. We added these new results and discuss a "single mechanism" interpretation in the discussion section, whereas we emphasize that such speculation needs further confirmation by future research.

We find a fixed effect of coupling on comprehension, with lower coupling being related to higher comprehension. Periodicity and syllabic rate both show a fixed effect after adjusting for the effect of coupling, with lower periodicity (higher MAD) and lower syllabic rates related to lower comprehension. Additionally, we find a moderating effect of the syllabic rate on the effect of coupling on comprehension (two-way interaction), periodicity moderated the effects of coupling on comprehension (two-way interaction). This was further specified by a 3-way interaction, suggesting that the relationship between comprehension and coupling is jointly moderated by syllabic rate and periodicity. Thereby, periodicity seems to particularly impact the moderating effect of syllabic rate on the comprehension-coupling relationship at slow syllabic rates (5 syl/sec, 11 syl/sec), with higher coupling related to higher comprehension for the high periodicity speech, whereas for the low periodicity speech no strong comprehension-coupling relationship seems present at slow syllabic rates. At higher syllabic rates for all periodicity conditions low coupling was related to high comprehension.

We revised the manuscript accordingly and discussed possible interpretations in the discussion section:

Abstract, l. 41: " Comprehension improved with lower periodicity and declined at faster rates, with the syllabic rate and periodicity moderating the coupling-comprehension relationship."

Introduction:

I. 117: “Auditory-motor coupling predicted comprehension in a complex relationship that was moderated by the syllabic rate and periodicity of speech. The findings are in line with a sweet spot for speech processing of natural, less periodic speech.”

Results:

I. 261: “In a direct analysis (GLMM) of the relationship of speech comprehension and auditory-motor coupling, we predict comprehension by the auditory-motor coupling (GCMI), the periodicity, the syllabic rate, as well as the interactions between coupling periodicity and syllabic rate. The interactions tests whether the relationship between comprehension and coupling (i.e. the slope) is moderated by the syllabic rate or periodicity. A fixed effect of GCMI ($b = 3.46$, $SE = 0.15$, $p < 0.001$), [...]”.

Discussion:

I. 288: “We observed a complex relationship between auditory-motor coupling and comprehension that was moderated by the syllabic rate and periodicity of speech. These results possibly reflect both: increased temporal predictions from the motor system to facilitate comprehension in demanding listening situations and optimal auditory cortex responsiveness to natural low periodicity speech.”

I. 349: “Interestingly, a direct analysis revealed a complex relationship between speech comprehension performance and auditory-motor coupling that was jointly moderated by the syllabic rate and the speech periodicity (Supplementary Figure 3). A fixed effect suggests that lower comprehension was related to higher auditory-motor coupling. This relationship, however, was further specified by two-way and three-way interactions. Both high and low auditory-motor coupling could predict high comprehension dependent on the periodicity and syllabic rate context. We speculate that the findings can be accounted for by a complex interplay of lower-level temporal processing in auditory cortex and higher-level temporal prediction from the speech motor cortices, possibly related to linguistic and prosodic information³³. More specifically, at slower syllabic rates, the motor cortex may provide temporal predictions of the speech signal. In the case of high periodicity speech such temporal motor predictions may be useful, as the periodic signal lacks the natural temporal structure of speech. This possibly resulted in higher auditory-motor coupling being related to higher comprehension when syllabic rates are slow and periodicity is high. In the case of low periodicity speech presented at slow syllabic rates, auditory cortex processing may be optimal -given that the natural temporal structure of speech is preserved- and individuals may more strongly rely on sensory representations. For fast syllabic rates given the demanding listening situation, temporal predictions from the motor cortex may be expected to be particularly useful, as shown for simple tone sequences⁸⁸. Others, however, did not find increased auditory-motor engagement at fast speech rates⁸⁵. We find that for fast speech rates higher comprehension was related to lower coupling. A possibility is that because of our speech production experience speech motor cortex provides exact temporal prediction for speech spoken at a natural slow tempo. At very fast rates outside of our production abilities, in contrast, predictions may be detrimental. Three-way interactions, however, should be interpreted cautiously and further research should explore this complex comprehension-coupling relationship. Furthermore, while the engagement of the motor system has been shown even during passive listening to speech^{38,39}, and with different tasks⁴¹, it is possible that our speech comprehension task emphasized the auditory-motor coupling, due to the requirement of verbally repeating the sentence.”

I. 394: “with a complex relationship between comprehension and coupling.”

Minor

1. In the introduction lines 90-97 a lot of concepts are introduced in quick succession; more unpacking of the literature can be done – especially how phase coupling in different bands is linked to temporal prediction (and at what level).

We restructured and revised the introduction to better explain our hypothesis, given the previous research.

I. 71: “Additionally, to such rhythm processing in auditory cortex, predictions from the motor cortex may be particularly relevant for processing the complex temporal structure of speech^{30–33}. Thereby, predictions may vary in strength for periodic and non-periodic speech, as features, such as prosody or higher-level linguistic processing affect the temporal predictability of speech^{7,11,12}. Interestingly, temporal predictions generated in speech-motor areas about upcoming sensory events during speech perception have also been linked to neural oscillations^{30,34–36}. Evidence for bi-directional theta-theta auditory-motor phase coupling has been provided^{37–39}, with the top-down coupling directly modulating the speech tracking in auditory cortex³⁹. Several spectral channels for predictive frontal-motor top-down effects during speech comprehension have been proposed: Two possibilities are phase coupling within the delta-theta band from frontal-motor to temporal cortex and between the beta and delta-theta band to primary auditory cortex. Effects often have been claimed in the delta band, but Park and colleagues suggested that it depends on the stimulus material whether effects in the delta or theta band are observed^{35,39} (see also: ³¹). Endogenous beta and delta-theta brain rhythms have been typically observed in speech motor cortices during rest^{40,41} and during language comprehension^{42–44}, possibly supporting timing and speech planning through coupling with the auditory cortex^{40,42,45}. While delta-theta to beta phase-amplitude coupling has been related to top-down temporal predictions from the speech motor cortex, delta-theta to gamma phase-amplitude coupling has been related to simpler bottom-up driven temporal predictions^{69,72,73}. In summary, according to the neural oscillatory account, we would expect higher speech periodicity to not only be beneficial for comprehension, but also modulate auditory-motor coupling.”

2. At the moment, the key manipulation of speech rate versus periodicity of stimuli is not explained very clearly in the introduction and only after reading Methods section the narrative becomes clear.

We now clarified this at the end of the introduction:

I. 112: “These manipulations allowed us to disentangle the effects of speech periodicity and syllabic rate, which are naturally intertwined. [...]!

3. Given the literature the authors review on the role of delta-theta coupling in temporal predictions, it is not clear why authors did not the conduct such analyses themselves? Especially, given their claim that more naturalistic and less periodic stimuli could produce greater temporal predictions that are not linked exclusively to syllabic level information.

See response to minor point 1 above:

We restructured and revised the introduction to better explain our hypothesis, given the previous research.

Dear Reviewers, many thanks for your continued time and effort and the constructive feedback. We have revised our manuscript according to the remaining reviewer suggestions.

In summary, we added a quantification of the transcription error based on additional raters. We added a novel Figure 3 that displays the results from the analysis of the relationship of the neural and comprehension data. The other figures were revised accordingly. All other comments have been responded to in the letter below.

Reviewers' comments:

Reviewer #1 (Remarks to the Author):

Thank you for this thoroughly revised manuscript. My concerns have been addressed.

Many thanks for your time and effort!

Reviewer #2 (Remarks to the Author):

I thank the authors for their thoughtful consideration of my and the other reviewers' comments. I am satisfied with the responses to my comments.

In particular, I agree that the beta models are a better strategy than binomial regression models.

Many thanks for your time and effort!

My remaining comments are:

Since the “word complexity” variable consists only of the average word frequency in the sentence, it should also be called something with frequency. “Complexity” in the context of sentences would indicate that other features (such as the number of nodes or clauses) plays into the measure, too.

This has been revised throughout the manuscript to call it “sentence-level average word-frequency”.

The transcription error would be relatively easy to quantify and I encourage the authors to add this, as it is good scientific practice. A quick way to do this would have another person check a random sample of 10% of the trials, listen to the audio recordings and compare the transcript. This can then be the basis of calculating inter-rater agreement.

We now quantified the transcription error by randomly selecting 5% of the behavioral responses (spoken sentences) from each participant at each syllabic rate. This resulted in 795 sentences that were double-transcribed. Specifically, the recordings of the spoken sentences were transcribed by two German native speaker research assistants (transcribing half of the sentences each). We decided to divide the work between two people to speed up the process and due to available capacity. (Note that due to a copying error and 1 month leave of one of

the first authors, data of 4/57 participants were not included in the transcription error computation. However, we deem 5% of 300 trials of each of the remaining 53 participants sufficient).

We computed a word overlap performance score. All text preprocessing and scoring were performed in Python (version 3.12.4), and word level comparisons were implemented using the *difflib* package. The overlap with the original transcription (mean: 0.897, standard error: 0.272) is in the range of what is typically reported (Mansfield et al., 2021, Interspeech), whereas the transcription error is expected to be higher for challenging speech.

We added this information to the manuscript like below:

I. 489: “**Behavioral Data Analysis.** In the context of Lubinus et al.⁹¹, the recorded participants' responses were manually transcribed by the co-first author to produce text excerpts. The responses were compared to the original sentences using Python's built-in sequence matcher algorithm from *difflib* package to measure comprehension accuracy. This algorithm evaluates the similarity between two sequences based on the order of their elements (words). The sequence matcher produced a similarity percentage for each sentence (word-overlap performance), which was used as the comprehension performance measure. To assess the transcription error, two native German speakers manually checked 5% of all trials (15 trials per participant, including 3 from each syllabic-rate condition; 795 trials in total) by listening to the recorded responses and transcribing them. These transcriptions showed 90% word-overlap with those produced by the co-author. The word-overlap score was computed in Python (version 3.12.4) using the *difflib* package.”

The pattern of results for the analysis of whether speech comprehension is predicted by auditory-motor coupling are quite complex. I can't find the information how the categorical predictors (syllable rate, periodicity) were coded. Only if a sum-to-zero coding scheme was used (e.g., sum coding) is it possible to interpret main effects in the presence of interaction effects. This information should be added to the manuscript (also for the other models).

Syllabic rate and periodicity were coded as metrical variable. Periodicity is computed as median absolute deviation (MAD) per trial, which is metrical. For the syllabic rate we had metrical values with one value per trial. The values varied around a mean value of the corresponding syllabic rate conditions (5 Syl/sec, 11 Syl/sec, 14 Syl/sec, 16 Syl/sec, 17.5 Syl/sec). Therefore, syllabic rate was also coded as metrical. The only variables that were coded as categorical was the region of interest (ROI) and the hemisphere. No sum-to-zero coding scheme was used here.

We clarified this in the text and removed the interpretation of the fixed effect of ROI:

I. 451: “This resulted in five syllabic rate conditions with the syllabic rate of trials varying around the mean syllabic rate of the condition (5 Syl/s, 11 Syl/s, 14 Syl/s, 16 Syl/s, 17.5 Syl/s) and with an overlapping distribution of compression factors (Figure 4c).”

I. 513: “Note that all variables were treated as metrical.”

I. 612: “Note that ROI and hemisphere were treated as categorical variables, while all other variables were treated as metric variables. Given that no sum-to-zero coding scheme was used, fixed effects of ROI and hemisphere in the presence of interaction effects need to be considered cautiously and therefore are not interpreted.”

Furthermore, we removed all interpretations of the direction of fixed effects in the presence of interaction effects from the discussion section. Note that this was only the case for the fixed effect of ROI and did not change our overall interpretation.

I. 39: “Faster syllabic rates and lower periodicity were associated with stronger coupling between the pSTG and inferior precentral gyrus, but also inferior frontal gyrus and supplementary motor areas.”

I. 293: “Although the auditory-motor coupling strength varied across brain areas, all regions showed sensitivity to periodicity.”

I. 332: “Notably, the overall coupling strength varied across ROI pairs.”

Why is the corresponding figure only in the supplemental material? I would find it nicer, also for the reader, to have this in the main text.

We revised the figures to integrate the results of the Behavior-Neural processing GLMM in the manuscript. Therefore, Fig. 2 and Fig. 3 have been merged and a new Fig. 3 has been added. See Fig. 2 and Fig. 3 pasted below.

Fig2.

Figure 2. Figure 2. Periodicity and Syllabic Rate affect Auditory-Motor Cortex Coupling (a-c), as further explored in a ROI specific analysis (d-f). a) The fixed effect of syllabic rate on coupling (normalized GCMi) observed in the GLMM is displayed. Coupling increased with higher syllabic rates. b) The figure shows the fixed effect of periodicity (MAD) on coupling. Coupling increased as MAD increased (indicating lower periodicity). c) The interaction between periodicity and ROI Pairs is shown. The iPCG-pSTG pair exhibits the highest GCMi increases with lower periodicity (higher MAD) followed by IFG-pSTG and SMA-pSTG. A positive effect of MAD reflects a negative effect of periodicity on neural coupling. Other interaction effects between syllabic rate, MAD, and brain area pairs were also evident, but are not displayed here. ROI specific analysis: d) A iPCG-pSTG GLMM analysis shows coupling (GCMi) strengthens with higher syllabic rates and greater MAD values (lower periodicity). A significant interaction effect between periodicity and syllabic rate is observed. e) IFG-pSTG GLMM analysis: Coupling increases with higher syllabic rates and lower periodicity. Reduced coupling is consistently seen at lower rates, with a significant interaction effect between periodicity and syllabic rate. f) SMA-pSTG GLMM analysis: GCMi rises with higher syllabic rates and lower periodicity. Lower rates are linked

to weaker coupling, and although the interaction effect is less pronounced, it remains statistically significant.

Fig3.

Figure 3. Speech periodicity and syllabic rate moderate the coupling-comprehension relationship. The figure displays how the three-way interaction between auditory-motor coupling (GCMi), syllabic rates and periodicity predicts speech comprehension. Lines within each panel represent quartiles of MAD, ranging from low MAD (high periodicity) to high MAD (low periodicity). At slower syllabic rates (5 Syl/s, 11 Syl/sec) for higher periodicity (low MAD) speech, higher auditory-motor coupling (GCMi) was related to higher comprehension. Less of a relationship seems evident for low periodicity speech (at 5 Syl/s). At higher syllabic rates higher comprehension was related to lower GCMi.

Regarding the amplification of motor area engagement by the sentence repetition task, I find the argument that a previous study by the authors did also find similar effects with a different task that didn't involve speaking. This should find it's way into the manuscript and not just in the response to the reviewers. This would also not let the limitation hanging like it is now.

Thank you for the feedback, we revised the manuscript to make this clearer.

I. 382: "It is possible that due to the requirement of verbally repeating the sentence, our speech comprehension task emphasized the auditory-motor coupling aspect. However, the engagement of the motor system has been shown even during passive listening to speech^{38,39}, and similar behavioral effects of the motor system on speech comprehension performance have been shown with tasks that do not involve overt speech production^{41,85}."

In the regression tables in the supplemental material, one of the columns is called "Statistics". What does it mean?

The column was revised, it reads now "z-values".

Is it really necessary to correct the p values with FDR? This is an honest question. Within the same model, one gets multiple p values, but they come from the same test, so to say, because they derive from the solution to the modelling process to which all predictors contributed jointly and at the same time. Across the different analyses reported in the experiment, different aspects of the collected data were analysed, so I am also not so sure about the necessity of the FDR correction. It definitely won't hurt, though.

Even within one mixed model, each p-value corresponds to a different null hypothesis. As the reviewer points out, the mixed model estimates all parameters jointly — accounting for shared variance and dependencies — but the *p-values* for different effects are still separate inferential tests. Joint estimation doesn't automatically adjust for how many hypotheses are tested or how they are interpreted later (Demirkale, Nettleton, Maity, 2010, Biometrics).

We therefore decided to keep the FDR correction and have added the reference in the manuscript, l. 600: "All statistical analyses were corrected for multiple comparisons using the false discovery rate (FDR) method ¹⁰⁶."

In the caption of each regression results table, please specify what model this was (e.g., beta model) and ideally add the formula.

We revised this in the manuscript:

l. 912: "The GLMM model is fitted with TMB using the beta family. The equation for this model is: Speech comprehension performance ~ syllabic rate * periodicity + compression factor + sentence-level average word-frequency + syllabus number + stimulation order + (1 | trials) + (1 | participants) + (0 + syllabic rate | subject)."

l. 920: "For this model, GLMM is fitted with a TMB using the beta family. The model equation is: Mutual information ~ syllabic rate³ * periodicity * Region of interest + compression factor + sentence-level average word-frequency + hemisphere + (1 | trials) + (1 | subject)."

l. 927: "Model fitting was performed using the TMB, specifically with the beta family. The used model equation is: Speech comprehension performance ~ GCMI * syllabic rate * periodicity + compression factor + sentence-level average word-frequency + syllabus number + stimulation order + (1 | trials) + (1 | participants).

Did all models converge without issues? I can see that the random effects often have very small values (close to 0). This can be an indication of convergence issues.

The models did not have any convergence issue.

Please add the tables as actual tables, not as screenshots.

This has been revised.

The readers would also profit from more human-readable predictor labels. E.g., poly(Syllabic rate, 3)²:MAD could easily be written as "Syllabic rate² x MAD" (with the ² meaning a superscript).

This has been revised throughout the manuscript.

l. 610: "A third-degree polynomial (indicated as *variable*³) [...]"

l. 619: "*Mutual information ~ syllabic rate³ * periodicity * Region of interest + compression factor + sentence-level average word-frequency + hemisphere + (1 | trials) + (1 | subject)*"

l. 622: "*Mutual information ~ syllabic rate³ * periodicity + compression factor + sentence-level average word-frequency + hemisphere + (1 | trials) + (1 | subject)*"

Reviewer #3 (Remarks to the Author):

My comments for the previous version of this paper have been sufficiently addressed by the authors, and the quality of the manuscript has been improved and merits publication. The additional analysis relating comprehension to the auditory-motor coupling clarified some of the key questions in the paper and could be better highlighted in the abstract.

Many thanks for your time and effort!

Thank you for this suggestion, we have revised the abstract accordingly:

I. 42: “The syllabic rate and periodicity moderated the coupling-comprehension relationship, possibly reflecting a complex interplay of lower-level auditory processing and higher-level prediction from the speech motor cortices.”